

# Assessment of wavelet-based spatial verification by means of a stochastic precipitation model (wv_verif v0.1.0)

Sebastian Buschow[1], Jakiw Pidstrigach[1], and Petra Friederichs[1]

[1]Institute of Geoscience and Meteorology, University of Bonn

**Correspondence:** Sebastian Buschow (sebastian.buschow@uni-bonn.de)

**Abstract.**

The quality of precipitation forecasts is difficult to evaluate objectively because images with disjoint features surrounded by zero intensities cannot easily be compared pixel by pixel: Any displacement between observed and predicted field is punished twice, generally leading to better marks for coarser models. To answer the question whether a highly resolved model truly

delivers an improved representation of precipitation processes, alternative tools are thus needed. Wavelet transformations can be used to summarize high-dimensional data in a few numbers which characterize the field's texture. A comparison of the transformed fields judges models solely based on their ability to predict spatial correlations. The fidelity of the forecast's overall structure is thus investigated separately from potential errors in feature location. This study introduces several new wavelet based structure-scores for the verification of deterministic as well as ensemble predictions. Their properties are rigorously

tested in an idealized setting: A recently developed stochastic model for precipitation extremes generates realistic pairs of synthetic observations and forecasts with prespecified spatial correlations. The wavelet-scores are found to react sensitively to differences in structural properties, meaning that the objectively best forecast can be determined even in cases where this task is difficult to accomplish by naked eye. Random rain fields prove to be a useful test-bed for any verification tool that aims for an assessment of structure.

## 1   Introduction

The verification of quantitative precipitation forecasts basically consists of comparing two or more images which contain regions of complicated structure, surrounded by blank space. When the typical size of individual precipitating objects is not far below the spatial resolution of forecasts and observations, a gridpoint-wise comparison is unhelpful: If a given rain field is forecast perfectly, but slightly displaced, point-wise verification will punish the error twice, once at the points where precip-

itation is missing and once at the points where it was erroneously placed. Following the advent of high-resolution numerical weather predictions, this effect, known as *double penalty* (Ebert, 2008), has motivated the introduction of numerous new spatial verification tools.

In a comprehensive review of the field, Gilleland et al. (2009) identified four main strategies to avoid the double penalty problem and supply useful diagnostic information on the nature and gravity of forecast errors. Authors such as Theis et al.

(2005) and Roberts and Lean (2008) attempt to ameliorate the issue via successive application of spatial smoothing filters.





Other research groups such as Keil and Craig (2009), Gilleland et al. (2010) and recently Han and Szunyogh (2018) explicitly measure and correct displacement errors by continuously deforming the forecast into the observed field. Another popular approach consists of automatically identifying discrete objects in each field and subsequently comparing the properties of these objects instead of the underlying fields. Examples from this category include the MODE technique of Davis et al. (2006)

as well as the SAL of Wernli et al. (2008).

The last class of spatial verification strategies contains so-called scale-separation techniques which employ some form of high- and low-pass filters to quantify errors on a hierarchy of scales. A classical example of this family is the wavelet-based intensity-scale-score of Casati et al. (2004), more recent representatives include Yano and Jakubiak (2016), Marzban and Sandgathe (2009), Scheuerer and Hamill (2015) and Ekström (2016). The common idea of the latter three papers is that errors,

which neither relate to the marginal distribution nor to the location of individual features, should manifest themselves in the field's spatial covariance matrix. Direct estimates of all covariances would require unrealistically large ensemble data-sets or restrictive distributional assumptions. The authors cited above therefore base their verification on the fields' variograms, which are directly related to the spatial auto-correlations (Bachmaier and Backes, 2011) but can be estimated from a single field under the assumption that pairwise differences between values at two grid-points only depend on the distance between those

locations. This is the so-called *intrinsic hypothesis* (Matheron, 1963). Similarly, one could require stationarity of the spatial correlations themselves, in which case the desired information is contained within the field's Fourier transform. Both of these stationarity-assumptions may be inadequate in realistic situations where, the structure of the data varies systematically across the domain, for example due to orographic forcing, the distribution of water bodies or persistent circulation features.

Weniger et al. (2017) have suggested an alternative approach based on wavelets. The key result in this context comes from

the field of texture analysis, where Eckley et al. (2010) proved that the output of a two dimensional discrete redundant wavelet transform (RDWT) is directly connected to the spatial covariances. The crucial advantage of their approach is that it merely requires the spatial variation of covariances to be *slow*, not zero.

After some initial experiments by Weniger et al. (2017), this framework has successfully been applied to the ensemble verification of quantitative precipitation forecasts by Kapp et al. (2018). Their methodology consists of 1) performing the

corrected RDWT, following Eckley et al. (2010), to obtain local wavelet spectra at all grid-points, 2) averaging these spectra over space, 3) reducing the dimension of these average spectra via linear discriminant analysis and 4) verifying the forecast via the logarithmic score.

In this study, we aim to expand on their pioneering work in several ways. Firstly, we argue that the aggregation method of simple spatial averaging is not the only sensible approach. An alternative is introduced which incidentally suggests a compact

way of visualizing the results of the RDWT: Instead of aggregating in the spatial domain, we first aggregate in the scale-domain by calculating the dominant scale at each location. Secondly, we use both kinds of spatial aggregates to introduce a series of new, wavelet-based scores. In contrast to Kapp et al. (2018), we consider both the ensemble case and the deterministic task of comparing individual fields and avoid the need for further data reduction. We furthermore demonstrate how to obtain a well defined sign for the error, indicating whether forecast fields are too small- or too large-scaled. The experiments performed to

study the properties of our scores constitute another main innovation: The recently developed stochastic rain model of Hewer



(2018) allows us to set up a controlled yet realistic test-bed, where the differences between synthetic forecasts and observations lie solely in the covariances and can be finely tuned at will. In contrast to similar tests performed by Marzban and Sandgathe (2009) and Scheuerer and Hamill (2015), our data is physically consistent and thus bears close resemblance to observed rain fields. Lastly, we consider the choice of mother-wavelet in detail, using the rigorous wavelet-selection procedure of Goel and

Vidakovic (1995). The sensitivity of all newly introduced scores to the wavelet-choice is assessed as well.

The remainder of this paper is structured as follows: The stochastic model of Hewer (2018) is introduced in section 2. Sections 3 and 4 discuss the wavelet transformation and spatial aggregation in detail. The general sensitivity of the wavelet spectra to changes in correlation structure is experimentally tested in section 5. Based on these results, we define several possible deterministic and probabilistic scores in section 6. In a second set of experiments (section 7), we simulate synthetic

sets of observations and predictions and test our scores' ability to correctly determine the best forecast. A comprehensive discussion of all results is given in section 8.

## 2 Data: Stochastic rain fields

In order to test whether our methodology can indeed detect structural differences between rain fields, we need a reasonably large rain-like data-set whose structure is, to some extent, known a priori. Faced with a similar task, Wernli et al. (2008),

Ahijevych et al. (2009) and others have employed purely geometric test cases. While those experiments are educational, we would argue that the simple, regular texture of such data bears too little resemblance with reality to constitute a sensible test-case for our purposes. As an alternative, Marzban and Sandgathe (2009) considered Gaussian random fields, which have the advantage that the texture is more interesting and can be changed continuously via the parameters of the correlation model. However, since precipitation is generally known to follow non-Gaussian distributions, the realism of this approach is arguably

still lacking.

In this study, we generate a more realistic testing environment using the work of Hewer (2018), who developed a physically consistent stochastic model of precipitation fields based on the moisture budget:

$$P = \max\left( E - T - \mathbf{v} \cdot \nabla q - q\nabla \cdot \mathbf{v}, \ 0 \right), \tag{1}$$

where $P$ denotes precipitation, $E$ is a constant evaporation rate (in practice set to zero without loss of generality), $q$ is the abso-

lute humidity and $\mathbf{v} = (u, v)^T$ is the horizontal wind field. The threshold $T$ is chosen such that a pre-specified percentage of the field has non-zero values. The velocity and its divergence are represented via the two-dimensional Helmholtz decomposition, which reads

$$\mathbf{v} = \nabla \times \Psi + \nabla \chi \quad \Rightarrow \quad \nabla \cdot \mathbf{v} = \nabla^2 \chi,$$

where $\nabla \times \Psi := (-\partial_x \Psi, \partial_y \Psi)^T$ is the rotation of the streamfunction and $\chi$ is the velocity potential. The spatial process of

$P$ is thus completely determined by $(\Psi, \chi, q)^T$, which we model as a multivariate Gaussian random field with zero mean and





**Figure 1.** Example realizations of the stochastic rain-model on a $128 \times 128$ grid for various choices of scale $b$ and smoothness $\nu$. The threshold $T$ was chosen such that 20 % of the field has non-zero values.





covariance matrix

$$\mathrm{Cov}\left((\Psi_{\mathbf{s}}, \chi_{\mathbf{s}}, q_{\mathbf{s}})^T, (\Psi_{\mathbf{t}}, \chi_{\mathbf{t}}, q_{\mathbf{t}})^T\right) = \Sigma_{\Psi, \chi, q} \cdot M\left(\,||b(\mathbf{t} - \mathbf{s})||, \nu\,\right). \tag{2}$$

Here, $\mathbf{t}, \mathbf{s} \in \mathbb{R}^2$ are two locations within the 2D-domain and $M$ is the Matérn covariance function. The parameter $b$ governs the scale of the correlations, the smoothness parameter $\nu$ determines the differentiability of the paths. The matrix $\Sigma_{\Psi, \chi, q}$ is set to

unity for our experiments, meaning that the velocity components and humidity are uncorrelated. Preliminary tests have shown that these parameters have negligible effects on the structural properties of the resulting rain fields. The covariances needed to simulate a realization of $P$ via Eq. (1), i.e.,

$$\mathrm{Cov}\left(\left[q_{\mathbf{s}}, \nabla \cdot q_{\mathbf{s}}, \nabla \chi_{\mathbf{s}} - \nabla \times \Psi_{\mathbf{s}}, \nabla^2 \chi_{\mathbf{s}}\right]^T, \left[q_{\mathbf{t}}, \nabla \cdot q_{\mathbf{t}}, \nabla \chi_{\mathbf{t}} - \nabla \times \Psi_{\mathbf{t}}, \nabla^2 \chi_{\mathbf{t}}\right]^T\right)$$

follow from Eq. (2) by taking the respective mean-square derivatives. In the special case where $\Psi$, $\chi$ and $q$ are uncorrelated,

these three Gaussian fields, as well as the necessary first and second derivatives can directly be simulated via the `RMcurlfree` model from the R-package `RandomFields` (Schlather et al., 2013). Since the paths of a Matérn-process are $\min\{n \in \mathbb{N} | n > \nu\} - 1$ times differentiable, the model is ill-defined for $\nu \leq 1$. While the underlying distributions of $\Psi$, $\chi$ and $q$ are Gaussian, the precipitation process, consisting of non-linear combinations of the derived fields, can exhibit non-Gaussian behavior. For further details, the reader is referred to Hewer et al. (2017), Hewer (2018) and references therein.

Fig. 1 shows several realizations of $P$. We recognize that the model produces realistic-looking rain-fields, at least for moderately low smoothness and large scales, i.e., small values of $\nu$ and $b$. Two important restrictions imposed by Eq. (2) become apparent as well: Firstly, the model is isotropic, meaning that it cannot produce the elongated, linear structures which are typical of frontal precipitation fields. Secondly, covariances are stationary, implying the same texture across the entire domain. An anisotropic extension of this model is theoretically relatively straightforward (replacing the scalar parameter $b$ by a rotation

matrix), but the technical implementation remains a non-trivial problem. The search for a non-stationary version is an open research question in its own right. Since we would nonetheless like to address the general case, we propose a pragmatic alternative, which allows for non-stationarity and anisotropy at the cost of also losing the dynamical consistency imposed by Eq. (1). To this end, let

$$P = \max(P' - T,\ 0), \qquad P' \sim \mathcal{N}\left(\mathbf{0},\ M\left[||(\mathbf{t} - \mathbf{s})||, \nu(x, y, \mathbf{b}, \theta)\right]\right), \tag{3}$$

$$\nu(x, y, \mathbf{b}, \theta) = \exp\left(-\left|\left|\begin{bmatrix} b_1 \cos(\theta) & -b_2 \sin(\theta) \\ b_1 \sin(\theta) & b_2 \cos(\theta) \end{bmatrix} \cdot \begin{bmatrix} x \\ y \end{bmatrix}\right|\right|\right) \tag{4}$$

Precipitation is thus given by a truncated Gaussian random field with Matérn-covariance, the smoothness of which varies across space. This model is implemented in the `RandomFields` R-package (Schlather et al., 2013) as well.

The example fields shown in Fig. 2 are visually not dissimilar from the physics-based test cases (Fig. 1), albeit with a clearly visible change in texture across the domain. Both models will be used in section 7. Throughout our experiments, we will

normalize all realizations of these synthetic rain fields to unit sum, thereby removing any differences in total intensity and allowing us to concentrate on structure alone.



## 3 The redundant discrete wavelet transform

The technical core of our methodology consist of projecting the fields onto a series of so-called daughter wavelets $\psi_{j,l,\mathbf{u}}(\mathbf{r})$ : $\mathbb{R}^2 \to \mathbb{R}$, which are all obtained from a mother-wavelet $\psi(\mathbf{r})$ via scaling by $\mathbf{r} \to \mathbf{r}/j$, a shift by $\mathbf{r} \to \mathbf{r} - \mathbf{u}$ and rotation in the direction denoted by $l$. We must therefore choose a mother $\psi$ and decide, which values of $\{j, l, \mathbf{u}\}$ to allow.

5  Starting with the latter decision and guided by our desire to capture the field's covariance structure, we follow Weniger et al. (2017) and Kapp et al. (2018) in choosing a redundant discrete wavelet transform (RDWT). In this framework, the shift $\mathbf{u}$ takes on all possible discrete values, meaning that the daughters are shifted to all locations on the discrete grid. The scale $j$ is restricted to powers of two and the daughters have three orientations with $l = 1, 2, 3$ denoting the horizontal, vertical and diagonal direction, respectively. The projection onto these daughter wavelets, for which efficient algorithms are implemented in

10 the R-package `wavethresh` (Nason, 2016), transforms a $2^J \times 2^J$ field into $3 \times J \times 2^J \times 2^J$ coefficients, one for each location, scale and direction. Our decision in favour of the RDWT is motivated by a relevant result proven in Eckley et al. (2010). Let

$$X(\mathbf{r}) = \sum_{\text{all } j,l,\mathbf{u}} \underbrace{W_{j,l,\mathbf{u}}}_{\text{weight}} \cdot \underbrace{\psi_{j,l,\mathbf{u}}(\mathbf{r})}_{\text{daughter}} \cdot \underbrace{\xi_{j,l,\mathbf{u}}}_{\text{noise}} \tag{5}$$

be the so-called *two-dimensional locally stationary wavelet process* (henceforth LS2W). The random increments $\xi_{j,l,\mathbf{u}}$ are assumed to be Gaussian white noise. *Local stationarity* loosely speaking means that $X$'s auto-correlation varies infinitely

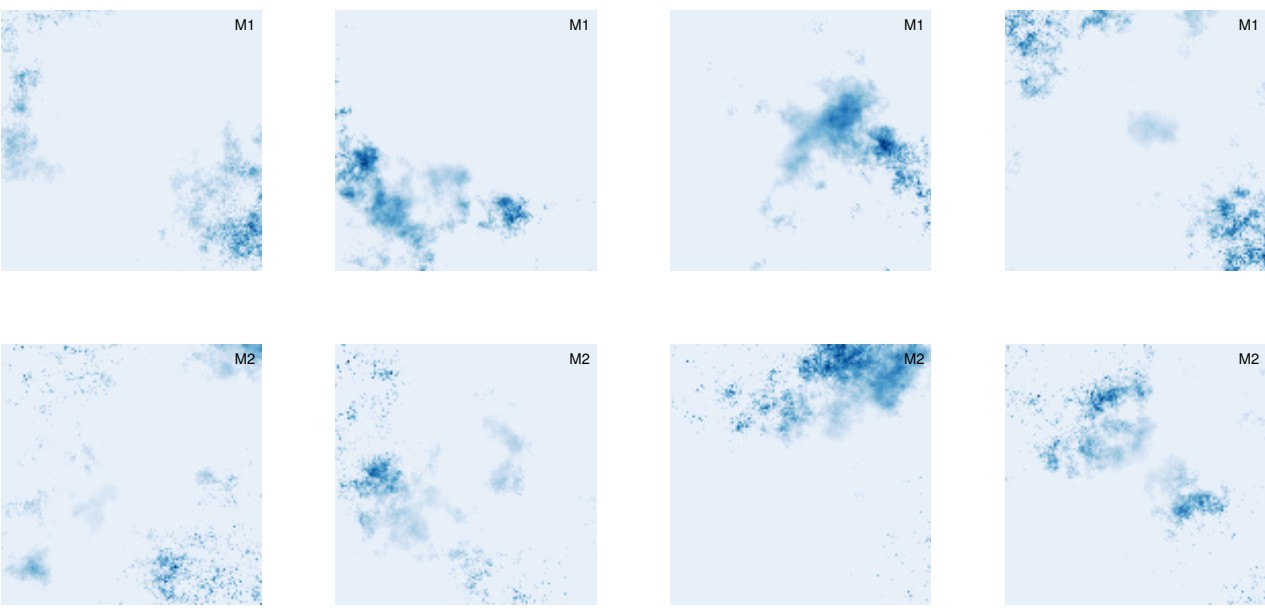

**Figure 2.** Example realisations of the thresholded non-stationary Gaussian model with $(b_1, b_2) = (0.5, 1.5)$ (top row, denoted M1) and $(b_1, b_2) = (0.5, 2.5)$ (bottom row, denoted M2). In both cases, $\theta = 45°$, $x, y \in [-1, 1]$ and $80\%$ of the fields have non-zero values.





slowly in the limit of infinitely large domains or, equivalently, infinitely highly resolved versions of a unit-sized domain. This requirement is enforced by certain asymptotic regularity conditions on the weights $W_{j,l,\mathbf{u}}$. For all technical details the reader is referred to Eckley et al. (2010), Kapp et al. (2018) present a more condensed summary. The main result of Eckley et al. (2010) states that, in the limit of an infinitely high spatial resolution, the autocovariances of $X$ can directly be inferred from the

squared weights $|W|^2$. These authors have furthermore proven that the squared coefficients of $X$'s RDWT constitute a biased estimator of these limit weights. The bias is corrected via multiplication by a matrix $\mathbf{A}^{-1}$ which contains the correlations between the $\psi_{j,j,\mathbf{u}}$ and thus depends only on the choice of $\psi$ and the size/resolution of the domain. Away from the asymptotic limit, this step occasionally introduces negative values to the spectra, which have no physical interpretation and pose some practical challenges in the subsequent steps. Preliminary investigations have shown that the abundance of this *negative energy*

sharply decreases with the smoothness of the wavelet $\psi$ and mostly averages out when mean spectra over the complete domain are considered. In a final step, the corrected local spectra need to be smoothed spatially in order to obtain a consistent estimate. The complete procedure, including the computationally expensive calculation of $\mathbf{A}^{-1}$, is implemented in the R-package LS2W (Eckley et al., 2011).

Having decided on a type of transformation, we must select a mother wavelet $\psi$. Our decision is restricted by the fact that

the results of Eckley et al. (2010) have only been proven for the family of orthogonal Daubechies-wavelets. These widely used functions, henceforth denoted $D_N$, have compact support in the spatial domain, increasing values of $N$ indicate larger support sizes as well as greater smoothness. Smoother and hence more wave-like basis functions with better frequency localization are thus also more spread out in space. $D_1$, the only family member which can be written in closed form, is widely known as the *Haar-wavelet* (Haar, 1910) and has been applied in several previous verification studies (Casati et al., 2004; Weniger et al.,

2017; Kapp et al., 2018). For $N > 3$, the constraints on smoothness and support length allow for multiple solutions, two of which are typically used: The *extremal phase* solutions are optimally concentrated near the origin of their support, while the *least asymmetric* versions have the greatest symmetry (Mallat, 1999). $D_{1,2,3}$ belong in both sub-families, wherever a distinction is needed, we will label the two branches of the family as *ExP* and *LeA*, respectively. Among these available mother wavelets, we seek the basis that most closely resembles the data, thus justifying the model given in Eq. (5). To this end, we follow Goel

and Vidakovic (1995) and rank wavelets by their ability to compress the original data: The sparser the representation in a given wavelet basis, the greater the similarity between basis functions and data. Relegating all details concerning this procedure to appendix A, we merely note that the structure of the rain field, determined by the parameters $b$ and $\nu$, has substantially more impact on the efficiency of the compression than the choice of wavelet. Overall, the least Asymmetric version of $D_4$ is most frequently selected as the best basis (28 % of cases), followed by $D_1$ and $D_2$ (21 % each). Unless otherwise noted, we

will therefore employ this version of $D_4$ in all subsequent experiments. Considering the relatively small differences between wavelets, we hypothesize that the basis-selection should have only minor effects on the behaviour of the resulting verification measures – a claim which is tested empirically in section 7.



## 4 Spatial aggregation

The previous step's redundant transform inflates the data by a factor of $3 \times J$, meaning that a radical dimension reduction is needed before verification can take place. Throughout this study, we will always begin this process by discarding the two largest scales, which are mostly determined by boundary conditions, and averaging over the three directions. The latter step is

unproblematic at least for the isotropic test cases – whether or not too much information is thus discarded in the anisotropic case will be seen in section 7. Next, the resulting fields must be spatially aggregated in a way that eliminates the double-penalty effect.

The straightforward approach to this task consists of simply averaging the wavelet spectra over all locations. The redundancy of the transform guarantees that this mean spectrum is invariant under shifts of the underlying field (Nason et al., 2000). Kapp

et al. (2018) have already demonstrated that the spatial mean contains enough information to confidently distinguish between weather situations in a realistic setting. In particular, the difference between organized large-scale precipitation and scattered convection has a clear signature in these spectra – an observation that has recently been exploited by Brune et al. (2018) who defined a series of wavelet-based indices of convective organization using this approach. Preliminary experiments have furthermore shown that *negative energy*, introduced by the correction matrix $\mathbf{A}^{-1}$, mostly averages out in the spatial mean,

provided that we choose a wavelet smoother than $D_1$.

In spite of these desirable properties, there are two main issues which motivate us to consider an alternative way of aggregation: If we normalize the mean spectrum to unit total energy, its individual values can be interpreted as the fraction of total *rain*

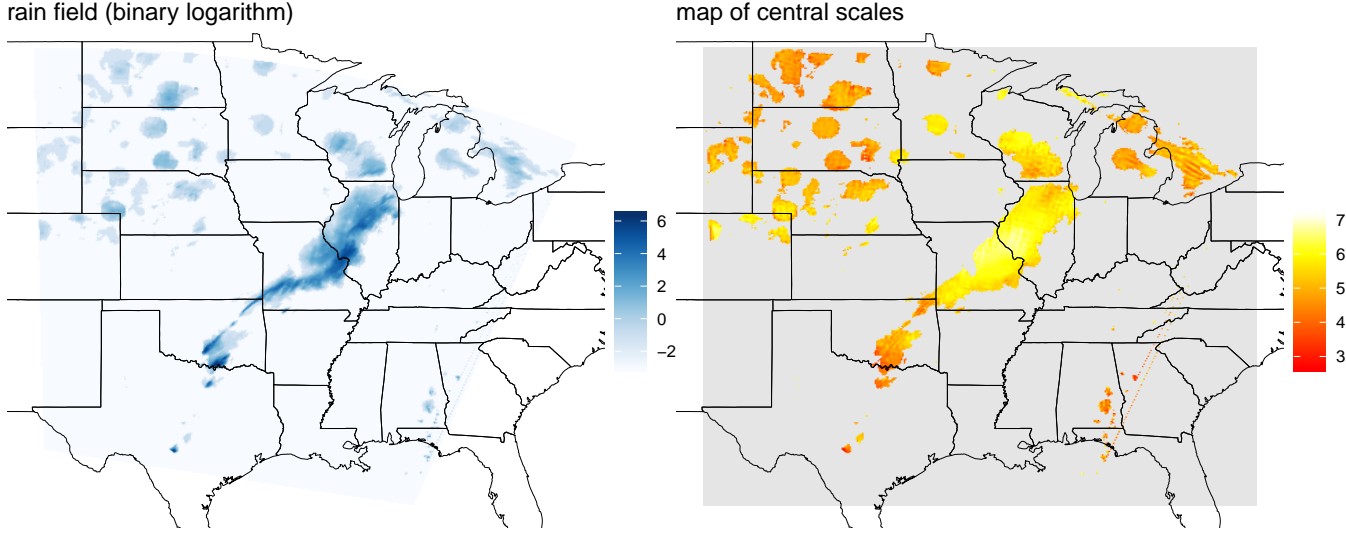

**Figure 3.** Logarithmized rain field and corresponding map of central scales from the stage II reanalysis on 26-04-2005 as used by Ahijevych et al. (2009) and contained in the `SpatialVx`-package. The field has been cut and padded with zeroes to $512 \times 512$, scales were calculated using the least asymmetric $D_4$ wavelet, only locations with non-zero rain are shown.





*intensity* associated with a given scale and direction. It is easy to imagine cases where a very small fraction of the total precipitation area contains almost all of the total intensity and therefore dominates the mean spectrum. This is clearly at odds with the intuitive concept of *texture*. Furthermore, there is no obvious way of visualizing how individual parts of the domain contribute to the mean spectrum – if our visual assessment disagrees with the wavelet-based score, we can hardly look at all fields of

coefficients at once in order to pinpoint the origin of the dispute. This second point leads us to introduce the *map of central scales C*: For every grid-point $(i,j)$ within the domain, we set $C_{i,j}$ to the centre of mass of the local wavelet spectrum. The resulting $2^J \times 2^J$ field of $C \in (1, J)$ is a straightforward visualization of the redundant wavelet transform, intuitively showing the dominant scale at each location. Since the centre of mass is only well-defined for non-negative vectors, all negative values introduced by the bias correction via $\mathsf{A}^{-1}$ are set to zero before computing $C$.

To illustrate the concept, we have calculated the map of central scales for one of the test cases from the `SpatialVx` R-package (Fig. 3). Here, the original rain field was logarithmized, adding $2^{-3}$ to all grid-points with zero rain. We see a clear distinction between the large frontal structure in the centre of the domain (scales 6-7), the medium-sized features in the upper left quadrant (scale 4-5) and the very small objects on the lower right (scales $\leq 4$ ).

As an alternative to the spatial mean spectrum, we can base our scores on the histogram of $C$ over all locations pooled

together. Intuitively, this scale-histogram summarizes which fraction of the total *area* is associated with features of various scales.

## 5   Wavelet spectra of random rain fields

Before we design verification tools based on the mean wavelet spectra and histograms of central scales, it is instructive to study what these curves look like and how they react to changes in the model parameters $b$, $\nu$ and $T$ from Eq. (1). Can we correctly

detect subtle differences in scale? What are the effects of smoothness and precipitation area? To answer these questions, we begin by simulating 100 realizations of the physically motivated model on a $128 \times 128$ grid, first keeping the smoothness $\nu$ constant at 2.5 and varying the scale $b$ between 0.1 and 0.5. For a second set of experiments, we simulate 100 fields with constant $b = 0.25$ and vary $\nu$ between 2.5 and 4. All of these fields are then normalized to unit sum (to eliminate differences in intensity), transformed and aggregated as described above.

Fig. 4 (a) shows the spatial mean spectra, averaged over all directions and realizations as a function of the scale parameter. As expected, an increase in $b$ monotonously shifts the centre of these spectra towards smaller scales. Simultaneously, the energy becomes less spread out, meaning that smaller scale-parameters $b$ lead to a greater variety of observed scales. Considering the experiment with variable $\nu$ (panel c), we find that an increase in smoothness results in a shift towards larger scales. This is in good agreement with the visual impression we get from the example realizations in Fig. 1. The effect on the spread of energy

across the spectra has the opposite sign here, meaning that smoother fields also feature a greater concentration of energy on a few scales.

The corresponding scale histograms are shown in Fig. 4 (b) and (d). We observe that their centres, corresponding to the expectation values of the central scales, are shifted in the same directions as the mean spectra. The spread of these distributions,





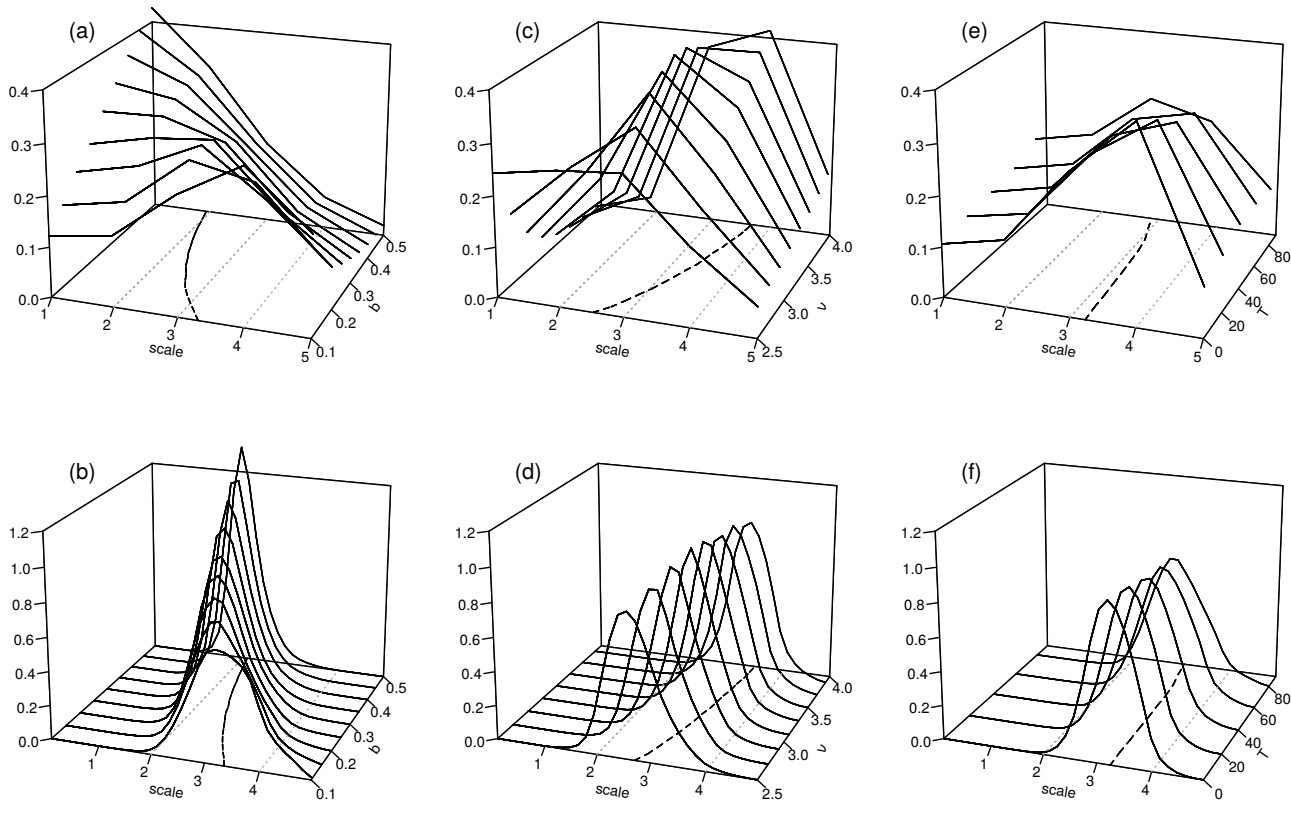

**Figure 4.** Mean spectra (top row) and histograms of central scale (bottom), as functions of the scale parameter $b$ at $\nu = 2.5$ (a,b), smoothness parameter $\nu$ at $b = 0.25$ (c,d) and threshold $T$ at $b = 0.1$, $\nu = 2.5$ (e,f). Dashed lines indicate the the respective curves' centres of mass, dotted gray lines (parallel to the y-axis) were added for orientation.

related to the variance of dominant scales across the domain, is however only affected by $b$: Smaller scale parameters lead to a greater variance, but changes in smoothness merely shift the histograms without substantially deforming them.

The final variable model parameter considered here is the threshold $T$, for which the expected reactions of our wavelet characteristics are less clear: Are fields with a larger fraction of precipitating area perceived to be larger- or smaller-scaled? To

5  investigate this, we set $b = 0.1$ and $\nu = 2.5$ and vary the rain-area between $10\,\%$ and $100\,\%$. Fig. 4 (e) and (f) show that the centres of the spectra and histograms hardly depend on $T$ at all. The spread slightly increases with the threshold in both cases, but the changes are far more subtle than for the other two parameters.

In summary, we note that the two structural parameters $\nu$ and $b$ have clearly visible effects on the mean spectra as well as the scale-histograms. Metrics that compare the complete curves (as opposed to their centres alone) should be able to distinguish

10  between errors in scale and smoothness since these characteristics have different effects on their location and spread. The effect





of the threshold $T$ is only moderate in comparison, but could potentially compensate errors in the other two parameters, which may occasionally lead to counterintuitive verdicts.

## 6   Wavlet-based scores

Motivated by the previous section's results, we now introduce several possible scores, comparing the spectra and histograms of forecast and observed rain fields. Here, we consider the case of a single deterministic prediction, as well as ensemble forecasts.

### 6.1   Deterministic setting

From an observed field and a single deterministic forecast, we obtain the respective mean wavelet spectra and histograms of central scales as described above. If we naively compare these vectors in an element-wise way, we may fall victim to a new incarnation of the double-penalty problem since a small shift in one of the spectra (or histograms) will indeed be punished twice. Rubner et al. (2000) discuss this issue in great detail and suggest the *earth mover's distance* (henceforth EMD) as an alternative. The EMD between two non-negative vectors (histograms or spectra in our case) is calculated by numerically minimizing the cost of transforming one vector into the other (*moving the dirt from one arrangement of piles to another while doing the minimal amount of work*). This quantity is a true metric if the two vectors have the same norm, which is trivially true for the histograms. To achieve the same for the mean spectra, we normalize them to unit sum. Our first two deterministic, wavelet-based structure scores are thus given by the EMD between the histograms of central scales (henceforth $H_{emd}$) and the normalized, spatially and directionally averaged wavelet spectra (henceforth $Sp_{emd}$), respectively.

Being a metric, the EMD is positive semi-definite and therefore yields no information on the direction of the error. We can obtain such a judgment by calculating, instead of the EMD, the difference between the respective centres of mass. For the histograms, this corresponds to the difference in expectation value. Rubner et al. (2000) have proven that the absolute value of this quantity is a lower bound of the EMD. Its sign indicates the direction in which the forecast spectrum or histogram is shifted, compared to the observations. We have thus obtained two additional scores, $H_{cd}$ and $Sp_{cd}$, which are conceptually and computationally simpler than the EMD-versions and allow us to decide whether the scales of the forecast fields are too large or too small.

### 6.2   Probabilistic setting

When predictions are made in the form of probability distributions (or samples from such a distribution), verification is typically performed using proper scoring rules (Gneiting and Raftery, 2007). Here, we treat scoring rules as cost functions to be minimized, meaning that low values indicate good forecasts. A function $\mathcal{S}$ that maps a probabilistic forecast and an observed event to the extended real line is then called a *proper* score when the predictive distribution $F$ minimizes the expected value of $\mathcal{S}$ as long as the observations are drawn from $F$. In this case, there is no incentive to predict anything other than one's best knowledge of the truth. $\mathcal{S}$ is called *strictly proper* when $F$ is the only forecast which attains that minimum. As mentioned above, Kapp et al. (2018) verified the spatial mean wavelet spectra via the logarithmic score, which necessitates a further di-





mension reduction step. In the interest of simplicity as well as consistency with our other scores, we employ the energy score (Gneiting and Raftery, 2007) instead, which is given by

$$\text{En}(F, \mathbf{y}) = E_F |\mathbf{X} - \mathbf{y}| - 0.5 E_F |\mathbf{X} - \mathbf{X}'|, \tag{6}$$

where $\mathbf{y}$ is the observed spectrum, $E_F$ denotes the expectation value under the distribution of the forecast spectrum $F$, and $\mathbf{X}$
and $\mathbf{X}'$ are independent random variables with distribution $F$. The resulting score, which we will denote as $\text{Sp}_e$, is proper in the sense that forecasters are encouraged to quote their true beliefs about the distribution of the spatial mean spectra.

The two previously introduced scores based on the histograms of central scales can directly be applied to the case of ensemble verification by estimating the forecast histogram from all ensemble members pooled together. In this setting where two distributions are compared directly, *proper divergences* (Thorarinsdottir et al., 2013) take the place of proper scores: A divergence, mapping predicted and observed distributions $F$ and $G$ to the real line, is called proper when its expected minimum lies at $F = G$. The square of $\text{H}_{cd}$ corresponds to the mean value divergence, which is proper. $\text{H}_{emd}$ is a special case of the Wasserstein distance, the propriety of which is only guaranteed in the limit of infinite sample sizes (Thorarinsdottir et al., 2013). Whether or not these divergences are useful verification tools in the probabilistic case will be tested empirically in section 7.

All of our newly proposed wavelet-based texture scores are listed in table 1.

**Table 1.** Wavelet-based structure-scores (top part) and established alternatives (bottom).

| abbreviation | description | probabilistic |
|---|---|---|
| $\text{Sp}_{emd}$ | EMD of the mean spectra | no |
| $\text{Sp}_{cd}$ | distance in mean spectra's centre of mass | no |
| $\text{H}_{emd}$ | EMD of the scale-histograms | yes |
| $\text{H}_{cd}$ | distance in the scale-histograms' centre of mass | yes |
| $\text{Sp}_e$ | energy score of the predicted mean spectra | yes |
| | | |
| RMSE | root mean square error between rain fields | no |
| $\text{V}_{w,5}$ | variogram score, $w_{i,j} = |\mathbf{r}_i - \mathbf{r}_j|^{-1}, p = 0.5$ | yes |
| $\text{V}_{20}$ | variogram score, $w_{i,j} = 1, p = 2$ | yes |
| S | object-based structure score of Wernli et al. (2008) | yes |





### 6.3 Established alternatives

In order to benchmark the performance of our new scores, we compare them to potential alternatives from the literature. A first natural choice is the variogram-score of Scheuerer and Hamill (2015), which is given by

$$V(F, \mathbf{y}) = \sum_{i,j=1}^{n} w_{i,j} \left( |y_i - y_j|^p - E_F[|X_i - X_j|^p] \right)^2 , \tag{7}$$

where $X$, $\mathbf{y}$ and $E_F$ are the same as in Eq. (6). The weights $w_{i,j}$ can be used to change the emphasis on pairs with small or large distances, while the exponent $p$ governs the relative importance of single extremely large differences. We include two versions of this score in our verification experiment: The naive choice $w_{i,j} = 1$, $p = 2$ (denoted $V_{20}$ below) and the more robust configuration $w_{i,j} = |\mathbf{r}_i - \mathbf{r}_j|^{-1}$, $p = 0.5$ ($V_{w,5}$ below), where $\mathbf{r}_i$ denotes the spatial location corresponding to the index $i$. Assuming stationarity of the data, we can efficiently calculate both of these scores by first aggregating the pairwise

differences over all pairs with the same distance in space up to a pre-selected maximum distance. $V_{20}$ then simplifies to the mean-square error between the two stationary variograms. The maximum distance is set to 20, which is a rough approximation of the range of the typical variograms of our test cases. Preliminary experiments have shown that this aggregation greatly improves the performances of $V_{w,5}$ and $V_{20}$ in all of our experiments. It furthermore allows us to apply these scores to the case of deterministic forecasts.

As a second alternative verification tool, we include the $S$-component of the well-known SAL-score (Wernli et al., 2008), which identifies continuous rain objects in the forecast and observed fields and subsequently compares the average shape and size of these objects. Here, we employ the original object identification algorithm of Wernli et al. (2008), setting the threshold to the maximum observed value divided by 15. We have checked that the sensitivity to this parameter is low in our test cases. For the purposes of ensemble verification, we employ a recently developed ensemble generalization of SAL (Radanovics et al.,

20 2018).

Lastly, the naive root mean square error (RMSE) will be included in our deterministic verification experiment in order to confirm the necessity for more sophisticated methods of analysis.

## 7 Idealized verification experiments

For our first set of randomly drawn forecasts and observations from the model given by Eq. (1), we keep the threshold $T$

constant such that 20 % of the fields have non-zero values and select four combinations of $\nu$ and $b$, listed in table 2. The

**Table 2.** Varying parameters in Eq. (2) for the four groups of artificial ensemble forecasts.

| model | RL | SmL | RS | SmS |
|---|---|---|---|---|
| smoothness $\nu$ | 2.5 | 3 | 2.5 | 3 |
| scale $b$ | 0.1 | 0.1 | 0.2 | 0.2 |





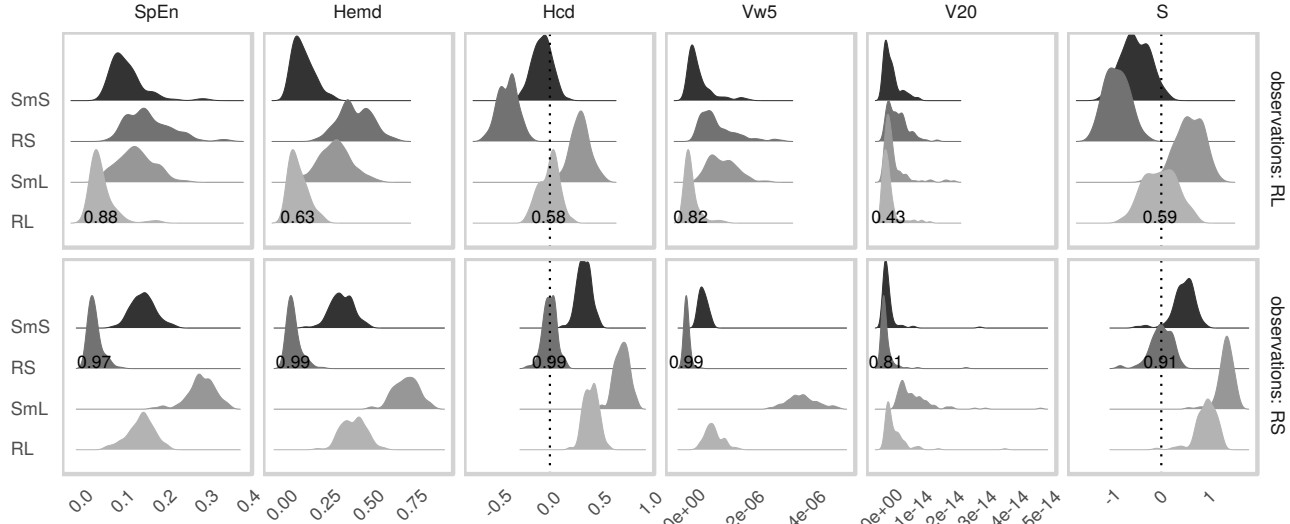

**Figure 5.** Distribution of all probabilistic scores for each of the forecast ensembles corresponding to the four models from table 2. Top row: Observations drawn from RL. Bottom: Observations drawn from RS. Numbers denote the fraction of cases where the forecast from the correct distribution received the best score.

resulting texture is rough and large-scaled (RL), smooth and large-scaled (SmL), rough and small-scaled (RS), and smooth small-scaled (SmS). One realization for each of those settings is depicted in Fig. 1. In the following sections, we interpret random samples of these models as observations and forecasts, thus allowing us to observe how frequently the truly best prediction (the one with the same parameters as the observation) is awarded the best score.

## 7.1 Ensemble setting

Beginning with the synthetic ensemble verification experiment, we draw 100 realizations each from RL and RS as our observations. For every observation (200 in total), we issue four ensemble predictions, consisting of ten realizations from RL, SmL, RS and SmS, respectively. Only one of these ten-member ensembles thus represents the correct correlation structure while the other three are wrong in either scale, smoothness or both. Observation and ensembles are compared via the three wavelet-scores $H_{cd}$, $H_{emd}$ and $Sp_e$ as well as the established alternatives S, $V_{20}$ and $V_{w,5}$.

Fig. 5 shows the resulting score-distributions. All scores are best when small, except for the two-sided S and $H_{cd}$ where values near zero are optimal. Beginning with the case where the observations are drawn from RS (bottom row of Fig. 5), we observe that the four predictions are ranked quite similarly by all scores. Here, the correct forecast almost always receives the best mark. S and $H_{cd}$ furthermore agree that all three false predictions are too large-scaled. The task of determining the truly best forecast is substantially more complicated when the observations belong to RL (upper row of Fig. 5): Since SmS is both smoother and smaller-scaled, the two errors have compensating effects on location of the spectra and histograms along the



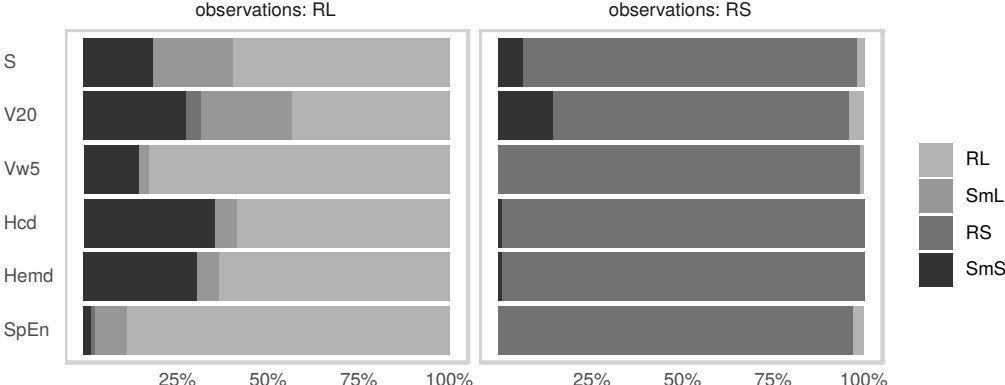

**Figure 6.** Percentage of cases where each of the four ensembles corresponding to the models in table 2 was deemed the best forecast, separated by score and the model of the observation.

scale-axis (cf. Fig. 4). These curves can therefore hardly be distinguished by their centres of mass alone. We recognize that RL and SmS consequently obtain similar values of $H_{cd}$, this score judging solely based on the centres. The other two wavelet-scores achieve better discrimination, as does $V_{w,5}$. Concerning the signs of the error, we note that S and $H_{cd}$ both consider RS too small and SmL too large. For SmS, $H_{cd}$ is close to zero, indicating approximately correct scales. S is less affected by the
compensating effect of increased smoothness and generally determines that SmS is smaller-scaled than RL. Its overall success rate is however not significantly better than that of $H_{cd}$.

Fig. 6 summarizes the ability of the six tested probabilistic scores to correctly determine the best forecast ensemble. As discussed above, all scores are very successful at determining correct forecasts of RS. In the alternative setting (observations from RL), SmS is the most frequent wrong answer, receiving the smallest (absolute) values of $V_{20}$, S, $H_{cd}$ and $H_{emd}$ in more
than a quarter of cases. In contrast to the other scores, $Sp_e$ hardly ever erroneously prefers SmS over RL. Instead, SmL is wrongly selected most frequently, leading to the overall lowest error rate (12 %) in this part of the experiment.

## 7.2    Deterministic setting

Having investigated the behaviour of our probabilistic scores, we now consider the deterministic case: How successfully can we determine the truly best forecast, given only a single realization? The set-up for this experiment is the same as before, only
the size of the forecast ensembles is reduced from ten to one. Since the resulting scores naturally have greater variances than before, we increase the number of observations to 1000 (500 each from RL and RS) in order to achieve similarly robust results. In addition to the four appropriate wavelet-scores ($Sp_{emd}$, $Sp_{cd}$, $H_{emd}$ and $H_{cd}$), we again calculate $V_{w,5}$ and $V_{20}$ as well as the S-component of the original SAL-score. To ensure that the verification problem is sufficiently difficult, the root mean square error (RMSE) is included as a naive alternative as well.



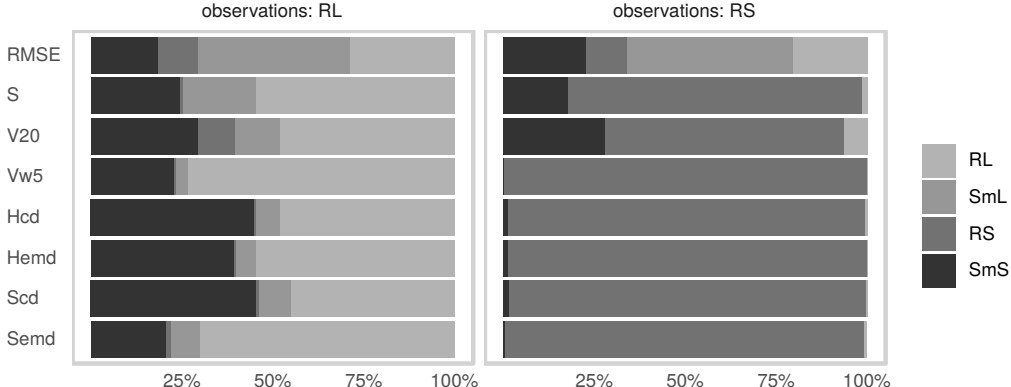

**Figure 7.** As Fig. 6, but for the deterministic verification experiment.

Fig. 7 reveals that correct forecasts are again easily identified by all of the wavelet-based scores when the observed fields belong to RS. As in the ensemble-scenario, the main difficulty lies in the decision between SmS and RL in cases where the latter model generates the observations. The two EMD-scores, which use the complete curves and not just their centres, clearly outperform the corresponding CD-versions in this part of the experiment and detect RL correctly in the majority of cases. $V_{w,5}$ is similarly successful as the best wavelet-based score, faring marginally better than $Sp_{emd}$. The failure rates of $V_{20}$ and S are again slightly higher. Unsurprisingly, the RMSE is completely unsuited to the task at hand, achieving less than 25 % correct verdicts overall. The inferiority to a random evaluation, which would, on average, be correct one fourth of the time, is caused by the fact that the model with the largest, smoothest features (SmL) has the least potential for double penalties and thus fares best in a point-wise comparison – in fact, RMSE orders the four models by their typical features size, irrespective of the distribution of the observation.

### 7.3 Wavelet choice and bias correction

One obvious question to ask is whether or not the choice of mother wavelet has a significant impact on the success rates in the two experiments discussed above. To address this issue, we repeat both the deterministic and the ensemble verification process for several Daubechies wavelets. Recalling the results of our objective wavelet selection (section 3 and appendix A), we expect no dramatic effects.

Table 3, listing the overall success rates for each tested wavelet, mostly confirms this expectation: In the deterministic case, $Sp_{emd}$ and $H_{emd}$ are really only affected by the choice between the Haar-wavelet, which performs worst, and any of its smoother cousins. The two centre-based scores ($Sp_{cd}$ and $H_{cd}$) show hardly any wavelet-dependence at all. Sensitivities are overall slightly higher in the ensemble case. While $D_1$ again appears to be the worst choice, there are some differences



**Table 3.** Fraction of cases where the correct forecast received the best score for a range of extremal phase (ExP) and least asymmetric (LeA) Daubechies wavelets.

|  | *deterministic case* | | | | *ensemble case* | | |
|---|---|---|---|---|---|---|---|
|  | $\mathrm{Sp}_{emd}$ | $\mathrm{Sp}_{cd}$ | $\mathrm{H}_{emd}$ | $\mathrm{H}_{cd}$ | $\mathrm{Sp}_{e}$ | $\mathrm{H}_{emd}$ | $\mathrm{H}_{cd}$ |
| ExP1 | 0.76 | 0.72 | 0.73 | 0.72 | 0.86 | 0.78 | 0.78 |
| ExP2 | 0.83 | 0.73 | 0.8 | 0.77 | 0.92 | 0.82 | 0.83 |
| ExP4 | 0.87 | 0.7 | 0.8 | 0.75 | 0.94 | 0.87 | 0.78 |
| ExP6 | 0.87 | 0.7 | 0.82 | 0.73 | 0.94 | 0.86 | 0.74 |
| LeA4 | 0.84 | 0.71 | 0.76 | 0.73 | 0.92 | 0.81 | 0.78 |
| LeA6 | 0.86 | 0.69 | 0.76 | 0.69 | 0.94 | 0.88 | 0.8 |

**Table 4.** As table 3, but without the bias-correction step.

|  | *deterministic case* | | | | *ensemble case* | | |
|---|---|---|---|---|---|---|---|
|  | $\mathrm{Sp}_{emd}$ | $\mathrm{Sp}_{cd}$ | $\mathrm{H}_{emd}$ | $\mathrm{H}_{cd}$ | $\mathrm{Sp}_{e}$ | $\mathrm{H}_{emd}$ | $\mathrm{H}_{cd}$ |
| ExP1 | 0.66 | 0.6 | 0.63 | 0.63 | 0.68 | 0.76 | 0.74 |
| ExP2 | 0.65 | 0.57 | 0.65 | 0.64 | 0.68 | 0.74 | 0.76 |
| ExP4 | 0.65 | 0.56 | 0.61 | 0.6 | 0.68 | 0.72 | 0.72 |
| ExP6 | 0.63 | 0.54 | 0.59 | 0.59 | 0.66 | 0.7 | 0.71 |
| LeA4 | 0.63 | 0.56 | 0.64 | 0.63 | 0.68 | 0.7 | 0.68 |
| LeA6 | 0.63 | 0.55 | 0.62 | 0.61 | 0.66 | 0.69 | 0.68 |

between the other options, particularly for the two histogram-scores. Generally speaking, the impacts of wavelet choice on our verification results are nonetheless rather limited, as long as the Haar wavelet is avoided.

To confirm that the bias correction following Eckley et al. (2010) is indeed a necessary part of our methodology, we repeat these experiments without applying the correction matrix $\mathbf{A}^{-1}$. Without discussing the details (table 4), we merely note that

5    the success rates decrease substantially (depending on score and wavelet), meaning that bias correction generally cannot be skipped.

## 7.4 Perturbed thresholds

Next, we consider the case where forecast and observations are subject to random perturbations which are not directly related to the underlying covariance model. One rather natural way of implementing this scenario consists of randomly perturbing the

10    thresholds, i.e., the fractions of the domain covered by non-zero precipitation. In a realistic context, such random differences between forecast and observation could be associated with a displacement error which shifts unduly large or small parts of a precipitation field into the forecast domain.



**Table 5.** Fraction of cases where the correct forecast received the best score. Top two rows: Deterministic forecasts with and without perturbed threshold. Middle: Ensemble forecasts with and without perturbed thresholds. Bottom: Non-stationary model (constant $T$, only ensemble verification).

|  |  | $\mathrm{Sp}_e$ | $\mathrm{Sp}_{emd}$ | $\mathrm{Sp}_{cd}$ | $\mathrm{H}_{emd}$ | $\mathrm{H}_{cd}$ | $\mathrm{V}_{w,5}$ | $\mathrm{V}_{20}$ | S | RMSE |
|---|---|---|---|---|---|---|---|---|---|---|
| *det.* | constant $T$ | - | 0.84 | 0.71 | 0.76 | 0.73 | 0.86 | 0.57 | 0.68 | 0.2 |
|  | random $T$ | - | 0.83 | 0.7 | 0.78 | 0.74 | 0.56 | 0.35 | 0.67 | 0.22 |
| *ens.* | constant $T$ | 0.92 | - | - | 0.81 | 0.78 | 0.9 | 0.62 | 0.75 | - |
|  | random $T$ | 0.92 | - | - | 0.8 | 0.75 | 0.7 | 0.44 | 0.74 | - |
| *non-stat. model* |  | 0.86 | - | - | 0.85 | 0.84 | 0.76 | 0.7 | 0.6 | - |

Our experiments in section 5 indicate that the wavelet-based scores should be relatively robust to small changes in the threshold $T$ (cf. Fig. 4 e and f). For the variogram-scores, one might expect greater sensitivity since the presence of a fixed fraction of zero-values greatly reduces the variance of the pairwise distances from which the stationary variogram is estimated. To test these hypotheses, we again repeat the two verification experiments, this time randomly varying $T$ such that the precipitation

area is a uniform random variable between 75 % and 85 % of complete domain.

Looking at the resulting success rates (table 5), we find our expectations largely confirmed: While variations in the precipitation coverage hardly influence our wavelet-based judgment, $\mathrm{V}_{w,5}$ and $\mathrm{V}_{20}$ seem to strongly depend on this parameter, thus mostly losing their ability to determine the correct model. The performances of S and RMSE are only weakly influenced by variations in $T$.

**7.5   The non-stationary case**

As discussed in section 2, the synthetic rain fields given by Eq. (1) are both stationary and isotropic. Since one major conceptual advantage of the wavelet-approach is its ability to deal with stochastic processes that fulfill neither of these assumptions, a more general test-case is of great interest. We therefore now consider the truncated Gaussian model given by Eq. (3). The angle of anisotropy is set to $\theta = 45°$, $x$ and $y$ vary in 128 steps between $-1$ and $1$, and $T$ is chosen such that 80 % of grid-points are

zero. This process produces large features near the centre of the domain, while spatial correlations decay increasingly quickly towards the edges. The large-scale region has an elliptical shape, the main axis of which is oriented along the field's diagonal (cf. contours of $\nu(x,y)$ shown in Fig. 8). The two free parameters $b_1$ and $b_2$, governing the ellipsis' aspect ratio, are used to set up our verification experiment. Here, we only consider the ensemble-case, letting our competing scores decide between two models: M1 with $(b_1, b_2) = (0.5, 1.5)$ and M2 with $(b_1, b_2) = (0.5, 2.5)$. One hundred realizations of each model serve as

"observations", the synthetic forecast-ensembles again consist of ten draws from M1 and M2 respectively. Considering the example fields shown in Fig. 2, we note that the distinction between the two models by naked eye is not trivial.



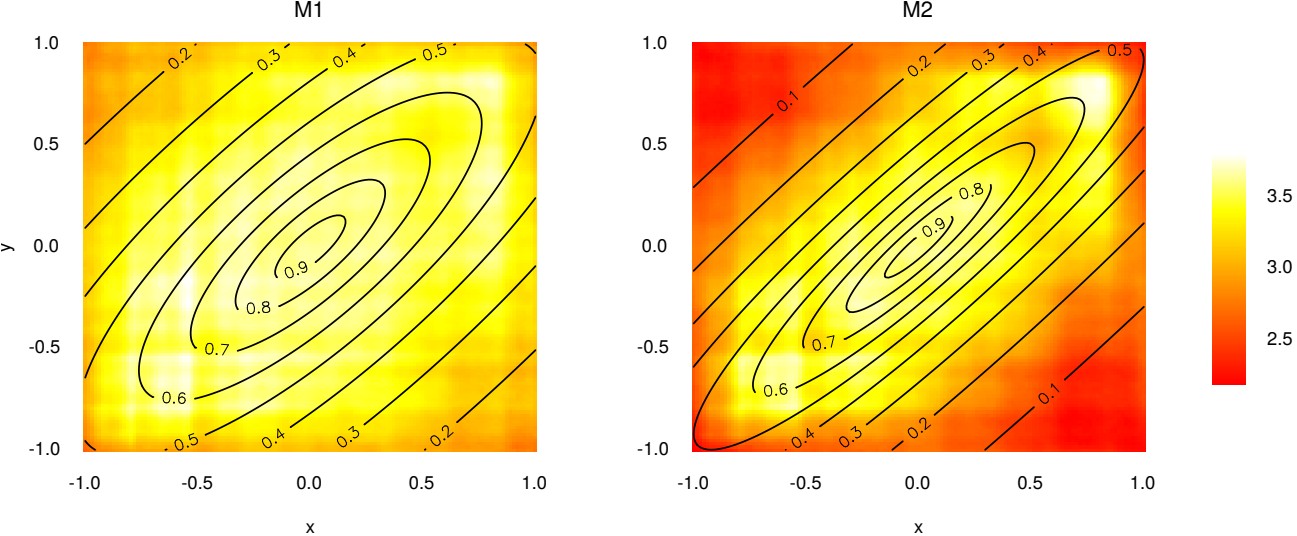

**Figure 8.** Contours of $\nu(x, y)$ and average map of central scales (color) for the two non-stationary models.

The success rates of the probabilistic scores (bottom part of table 5), confirm that this task is overall more difficult than the stationary examples considered before – especially keeping in mind that we provided only one false alternative. Somewhat surprisingly, the performance of the variogram scores, which explicitly rely on the now invalid assumption of stationarity, is still adequate. The wavelet-based scores, which do not explicitly require stationarity, perform substantially better in this setting. In particular, $H_{emd}$ and $H_{cd}$ which rely not on the spatial average but on the distribution of scales across the domain, are now on par with $Sp_e$.

Fig. 8 shows the map of central scales on which the histogram-scores are based, averaged over all realizations of the two models. We recognize that this "scale-climatology" nicely re-traces the shape and orientation of the underlying non-stationary smoothness function $\nu$ – in fact, the correlation between the mean map of scales and the field of $\nu$ is greater than 0.75 in both cases.

## 8 Discussion

The basic idea of this study is that the structure of precipitation fields can be isolated and subsequently compared using two-dimensional wavelet transforms. Building on the work of Eckley et al. (2010) and Kapp et al. (2018), we have argued that the corrected, smoothed version of the redundant discrete wavelet transform (RDWT) is an appropriate tool for this task since it is shift-invariant and has a proven asymptotic connection with the correlation function of the underlying spatial process. This approach is theoretically more flexible than Fourier- or variogram-based methods which make some form of global stationarity assumption, while our method relies on the substantially weaker requirement of local stationarity.





Before wavelet-transformed forecasts and observations can be compared to one another, the spatial data must be aggregated in a way that avoids penalizing displacement errors twice. Besides the proven strategy (Kapp et al., 2018) of averaging the wavelet-spectra over all locations, we have newly introduced the map of central scales as a potentially interesting alternative: By calculating the centre of mass for each local spectrum, we obtain a matrix of the same dimensions as the original field, each

value quantifying the locally dominant scale. Aside from the possibility of compactly visualizing the output of the RDWT in a single image, the histogram of these scales can serve as an alternative basis for verification, emphasizing each scale based on the area in which it dominates, rather than the fraction of total rain intensity it represents.

In order to rigorously test the sensitivity of these aggregated wavelet transforms to changes in the structure of rain fields, a controlled but realistic test-bed was needed. The stochastic precipitation model of Hewer (2018) constitutes a very convenient

case study for our purposes: The construction based on the moisture budget and a Helmholtz-decomposed wind-field allows for non-Gaussian behaviour and guarantees that the simulated data is more realistic than simple geometric patterns or Gaussian random fields. The model's structural properties can nonetheless be determined at will via the smoothness and scale parameter of the underlying Matérn fields, allowing us to simulate observations and forecasts with known error characteristics. In a realistic context, errors in scale correspond to mis-forecast feature sizes (missing fronts, underestimation of convective organization)

while errors in smoothness correspond to forecast models with too-coarse resolution, which are incapable of reproducing fine structures.

In a first suite of experiments we found that the wavelet-spectra do indeed react sensitively to changes in both of these parameters. In particular, errors in smoothness and scale have different signatures which can potentially be differentiated from one another. Encouraged by these results, we have defined several possible scores, which compare mean spectra and

scale-histograms via the difference of their centres ($H_{cd}$ and $Sp_{cd}$), their earth mover's distance ($H_{emd}$ and $Sp_{emd}$), and the energy score ($Sp_e$). In our idealized verification experiments, the performance of the latter three scores, i.e., their ability to correctly determine the objectively best forecast, was on par with the best tested variogram-score ($V_{w,5}$). The less robust $V_{20}$ as well as the SAL's structure component S and the simplistic RMSE were clearly out-performed. $H_{cd}$ and $Sp_{cd}$, while less proficient at finding the correct answer, do yield valuable auxiliary information in the form of the error's sign, answering the

question whether the predicted structure was too coarse or too fine. All five wavelet-scores were shown to be robust to small perturbations of the data, realized here as random changes to the fraction of non-zero rain. In this part of the experiments, the variograms scores largely lost their ability to determine the correct forecast. Interpreting this result, it is important to keep in mind that our wavelet-scores were specifically designed to judge based on structure alone while the variogram-methodology of Scheuerer and Hamill (2015) allows for a more holistic assessment. Sensitivity to precipitation coverage is therefore not

necessarily a disadvantage.

The variogram-score's two free parameters, namely the exponent $p$ and the choice of weights $w_{i,j}$, were found to have a significant impact on the resulting verification. With our wavelet approach, the potentially most important degree of freedom is the choice of mother wavelet. To address this issue, we have performed an objective wavelet selection procedure following Goel and Vidakovic (1995) and repeated the verification experiments for a variety of possible choices. Summarizing both

of these steps, we can conclude that the success of our wavelet-based verification depends only weakly on the choice of an



appropriate mother wavelet. One somewhat surprising exception is the Haar wavelet, which was favoured by previous studies (cf. Weniger et al. (2017) and references therein) but turned out to be a sub-optimal choice for our purposes.

In a final step, the physically motivated model of Hewer (2018) was replaced by truncated Gaussian random fields with non-stationary, anisotropic covariances. While some realism is thereby abandoned, the more general case is nonetheless important

since real rain fields are typically neither stationary nor isotropic. In this experiment, scores based on the newly introduced map of central scales, which arguably make better use of the wavelet's localization capabilities, were amongst the best tested verification tools, clearly out-performing the stationary variogram scores. The average map of scales itself was furthermore found to closely resemble the spatially varying smoothness function of the correlation models.

Now that the merits of wavelet-based structure scores have been demonstrated in a controlled environment, they are ready for

further application to real-world verification problems. One important open question concerns the use of direction information, which was neglected in the present study but may well be valuable in a more realistic scenario. It is furthermore worth noting that, in contrast to primarily rain-specific tools like SAL, our methodology can be applied to any variable of interest with no major changes besides the new selection of an appropriate mother wavelet. A simultaneous evaluation of, for example, wind components, humidity and cloud-cover, using the exact same verification tool to assess structural agreement in each variable,

is thus feasible and could answer interesting questions concerning the origins of specific systematic forecast deficiencies.

*Code and data availability.* All necessary R-code for the simulation of the stochastic rain fields, as well as the wavelet-based forecast verification, is available from https://github.com/s6sebusc/wv_verif.

## Appendix A: Entropy-based wavelet selection

To find the most appropriate wavelet, we calculate the entropy of the transform's squared coefficients (representing the energy

of the transformed data) and select the wavelet with the smallest entropy. Let $\mathbf{y} = (y_1, \ldots, y_n)^T$ be a vector with non-negative entries satisfying $\sum_i y_i = 1$. For our purposes, its entropy is defined as

$$s(\mathbf{y}) := -\sum_{i=1}^{n} y_i \log_2 y_i \quad \in \quad [0, \log_2(n)], \tag{A1}$$

where we set $0 \cdot \log_2(0) = 0$. Following Goel and Vidakovic (1995), the RDWT is replaced by its corresponding orthogonal decomposition, which is obtained by selecting every second of the finest-scale coefficients, every fourth on the second-finest

scale and so on. The number of data-points is thus conserved under the transformation and we can compare the entropy of the transformed data to that of the original representation.

The outcome of this procedure depends on the structure of the data to be transformed, the smoothness of the wavelet and the length of its support. To understand how these properties interact, we quantify smoothness via the number of vanishing moments: A wavelet $\psi$ is said to have $N$ vanishing moments if $\int x^q \psi(x) dx = 0$ for $q = 0, \ldots, N-1$. This implies that poly-

nomials of order $N-1$ have a very sparse representation in the wavelet-basis corresponding to $\psi$. The theorem of Deny-Lions



(Cohen, 2003) relates this property to a function's differentiability: Loosely speaking, if $f$ is $N$ times differentiable, the error made when approximating $f$ by polynomials of order $N-1$ is bounded by a constant times the energy of $f$'s $N$-th derivative $f^{(N)}$. It follows that $f$ is well represented by wavelets with $N$ vanishing moments, as long as $f^{(N)}$ is not too large.

Besides more or less smooth regions within the rain fields (in our test cases governed by the parameter $\nu$) and constant zero areas outside, the data we wish to transform also contains singularities at the edges of precipitating features. Here, $f^{(N)}$ is generally not small and wavelets with shorter support length are superior since fewer coefficients are affected by any given singularity. Heisenberg's uncertainty principle ensures that localization in space and approximation of polynomials (related to the localization in frequency) cannot both be optimal simultaneously: If a wavelet has $N$ vanishing moments, then its support size (in one dimension) is at least $2N-1$. In proving this theorem, Daubechies (1988) introduced the $D_N$-wavelets, which are optimal in the sense that they have $N$ vanishing moments at the smallest possible support.

To illustrate the competing effects of support size and smoothness on the efficiency of the wavelet transformation, we simulate one-dimensional Gaussian random fields with Matérn-covariances (same function $M$ and parameters $b$, $\nu$ as in Eq. (2), but only one variable and one spatial dimension). Fig. A1 neatly demonstrates the concepts discussed above: When the time-series is uniformly smooth, the higher order wavelet $D_4$ delivers a far more efficient compression than $D_1$ (panels a, c, e). The situation changes when we truncate the data (b, d, f): While $D_4$ continues to be superior within the smooth regions, $D_1$, due to its shorter support, requires fewer coefficients to represent the regions of constant zero values. This trade-off between representing smooth internal structure and intermittency is precisely quantified by the entropy (defined in Eq. (A1), values noted in the captions of Fig. A1 ), which measures the total degree of concentration on a small number of coefficients: While the $D_4$ does better in both cases, the relative and absolute improvement is worse in the cut off case, where we introduced artificial singularities.

Fig. A2 shows the results of our entropy-based wavelet selection procedure for the model given by Eq. (1). We observe that the model parameters have substantially more impact on the efficiency of the compression than the choice of wavelet. Fields with greater smoothness and larger scales (large values of $\nu$ and small values of $b$) are represented far more compactly than rough, small-scale cases, irrespective of the chosen basis. The differences between wavelets, while small in comparison, reveal a systematic behavior: Increasing support length leads to monotonously worse compression and the least asymmetric wavelets tend to fit slightly better than their 'extremal phase' counterparts. The Haar-wavelet constitutes an exception to this pattern, its entropy being frequently larger than that of several of its smoother cousins.

*Author contributions.* SB and PF developed the basic concept and methodology for this work. SB designed and carried out the experiments used to test the methods. JP investigated and carried out the wavelet selection procedure and led the writing and visualization in this part of the study. The rest of the writing and coding was led by SB with contributions from PF and JP. All authors contributed to the proof-reading and added valuable suggestions to the final draft.





*Competing interests.* The authors declare that they have no conflict of interest.

*Acknowledgements.* We are very grateful to Rüdiger Hewer for providing the original program code, as well as invaluable guidance, for the
stochastic rain model. Further thanks go to Franka Nawrath for providing efficient code to calculate the variogram-score, as well as helpful
discussions concerning its implementation. We would also like to thank Michael Weniger for many helpful suggestions concerning the use
5   of wavelets for forecast verification.

(a) time series, $s = 11.6$

(b) time series, $s = 10$

(c) $D_1$ transform, $s = 3.3$

(d) $D_1$ transform, $s = 3.6$

(e) $D_4$ transform, $s = 2.6$

(f) $D_4$ transform, $s = 3.1$

**Figure A1.** Realization of a one-dimensional Gaussian random vector with covariance $M(\nu = 3.5, b = 2)$ (a) and the corresponding values
of the $D_1$- and least asymmetric $D_4$-transform (c and e) which are greater than $0.1$. (b), (d) and (f) are the corresponding plots for the case
where the vector is cut off at zero.





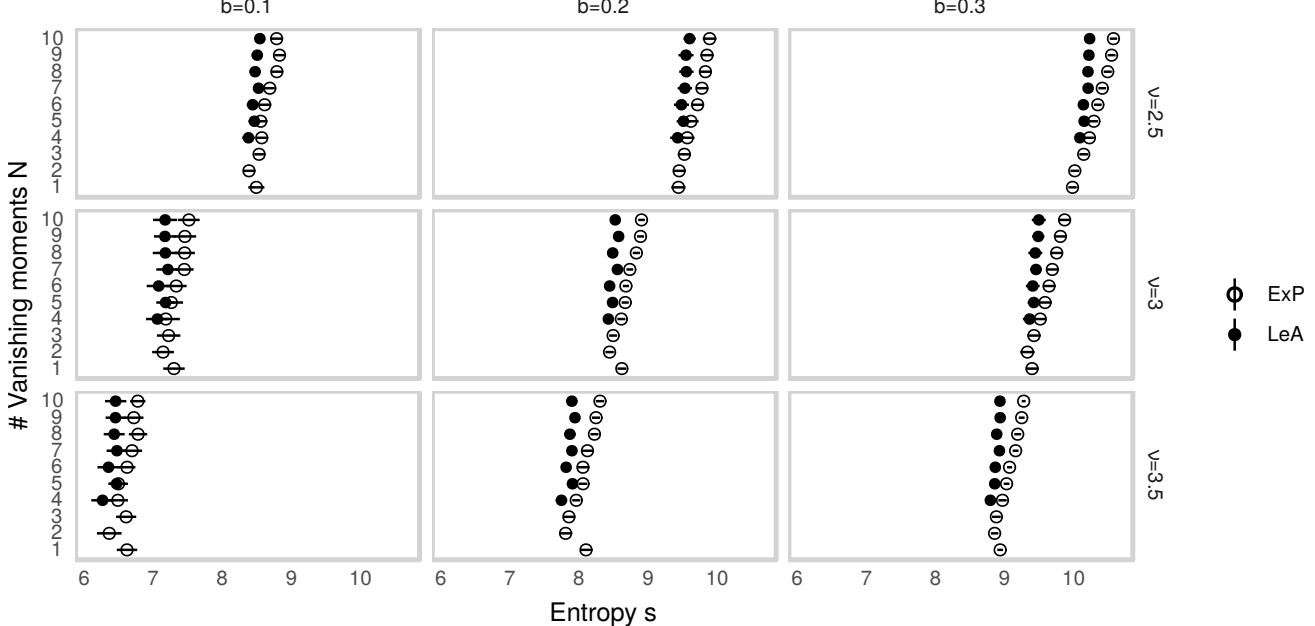

**Figure A2.** Entropy of the wavelet-transformed synthetic rain fields from Fig. 1 as a function of the wavelet's order $N$. Empty and filled dots correspond to the extremal phase- and least asymmetric versions of $D_N$. Lines indicate one standard deviation, estimated from ten realizations.

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
