# Peer review of "Assessment of wavelet-based spatial verification by means of a stochastic precipitation model (wv\_verif v0.1.0)"

_Geoscientific Model Development, 2019_

## Short Comment (SC1) · 15 May 2019

I am writing as an executive editor of GMD to highlight an issue with the code availability section which needs to be remedied in the revised manuscript.

Thank you for providing a reference to the full code and data used in the experiments presented in your manuscript. There are two problems with providing this data via GitHub. The first is that a reader cannot identify the exact version of the code that was used in the paper (for example, you may fix bugs or add features in the future). The second issue is that projects sometimes change the revision control system they use, or the hosting (the project might move to GitLab, for example). The solution to both of

these issues is to provide a reference to a persistent archive of the exact version of the code that was used in the manuscript. This reference can, and should, be in addition to the GitHub link, so that a user can also always access the most recent version of the code.

Since your original code is hosted on GitHub, the easiest way to produce a persistent archive of a precise version is to use GitHub's Zenodo integration. For more details, see: https://guides.github.com/activities/citable-code/.

Please ensure that the revised version of your manuscript contains a reference to a persistent, public archive of the exact version of the code used to produce it.
* * *

---

## Referee Comment (RC1) · Anonymous Referee #1 · 24 May 2019

Overall comments:

In this study five newly wavelet-based scores for assessing the forecast-versus-observation scale-structure are introduced, and their discriminatory characteristics are compared with that of other established (spatial) verification scores.

The article is well structured and the analysis is clearly explained. The set up for the scores definition and testing are well motivated (construction of the synthetic fields, in Sec 2; use of the redundant discrete wavelet transform and wavelet family selection, in Sec 3). Innovative parts of the article include: the spatial aggregation by considering the histogram of the dominant central scales (Sec 4); the analysis of the sensitivity

of the wavelet spectra to the scale and smoothness parameter (Sec 5); the use of the Earth Mover's Distance to compare the spectra (Sec 6). The set-up of the forecast experiment is ingenious (Sec 7) and enable a clear analysis of the discriminating power of the different scores (Fig. 6,7). Unfortunately, there is no single score which emerges as the recommended best score. However, the analysis is very solid and the methods illustrated are extremely interesting, therefore we recommend the article for publication (after minor-to-major revisions), as a very valuable contribution towards a better understanding (and development) of new spatial (scale-separation) verification methods.

Major Revisions:

It would be nice to see the mean spectra and histogram of the dominant scales for the case study shown in Figure 3 (please add a panel): I expect a bimodal histogram and spectra, since both small scale features and a large front are present in the case study. This bi-modality (i.e. presence of of both small and large scale features) is not represented in the stochastic rain fields produced in Section 2 (the synthetic fields considered in the article have by construction uni-modal spectra), and in fact their resulting spectra and histograms are uni-modal (e.g. Figure 4). However spectra bi-modality (i.e. presence of of both small and large scale features) is bound to happen in real verification practice, and it might be badly handled by H_cd and Sp_cd. In fact, H_cd and Sp_cd are the differences of the centre of mass (of the scale histograms and of the mean spectra), and are not suitable summary statistics to compare bimodal curves (or any other non-Gaussian curve). Hemd and Spemd, on the other hand, seem more suitable statistics (to compare Gaussian or non-Gaussian curves) since based on the whole curve comparison. The authors should consider withdrawing H_cd and Sp_cd from the newly proposed wavelet-based scores. [If the authors wish to introduce a metric which measure the direction of the error, maybe they should consider a measure based on the distances along the whole curves (or the integral between the two pdf), but with a sign which accounts for the curves relative position.]

Interactive
comment

Figure 4 (Section 5) shows that both mean spectra and scale histograms are sensitive to the variation of the scale parameter b and the smoothness parameter $\nu$, and that for both parameters, the curves shift in the expected direction (this is the main result). The histogram of the dominant scales seems slightly less sensitive (it shifts less), however it exhibits a smaller spreadÂă(hence smaller uncertaintyÂă: the signal is better defined). Because of this latter property, the scale histogram should be favoured, with respect to the mean spectra. Moreover, the smaller shifts of the histograms are probably simply related / due to their smaller spread (I have the feeling that the magnitude of the shift is proportional to the spread). These aspects should be mentioned in Section 5. (Note: the sensitivity of the spread to the parameters b and $\nu$ is secondary: be careful not to mix it up with the main result, aka the shift).

From the previous two comments, I would propose as unique new statistics H_emd.

At the end of Section 2, then authors introduce an algorithm for producing stochastic rain fields which satisfy non-stationarity and anisotropy. Some case studies are illustrated in Figure 2, and the associated verification results are discussed in Section 7.4. In my view this analysis can be removed from the article for the following reason:

a) The algorithm for producing stochastic rain fields which satisfy non-stationarity and anisotropy, despite being more sophisticated than the isotropic algorithm mainly used in the article, is still not realistic (the precipitation features of Figure 2 are still far from resembling the ones for the real case illustrated in Figure 3).

b) The article will result nicely well contained in illustrating "solely" the isotropic stochastic fields (you have already quite a lot of material! Moreover, this would provide a nice "excuse" for retaining the statistics based on the centres of mass -wink!-). In this case you need to add into the final discussion Section the need to analyze real cases, in future work ...

c) For the (future) analysis of more realistic cases, I strongly suggest to consider directly real precipitation case studies (the Spatial Verification ICP cases from Ahijevych

et al 2009 are available online), rather than using synthetic fields (you might end up spending a lot of time and implementing very complex stochastic models ... to achive the same results ... ).

Minor Revisions:

Abstract and Introduction

Page 1 line 7: replace 'spatial correlation' with 'spatial structure' (or 'scale structure'). Page 1, line 23: please quote (also) Dorninger et al (2018): "The set-up of the Mesoscale Verification Inter-Comparison over Complex Terrain project". Bull. Amer. Meteorol. Soc., 99 (9), 1887 – 1906. Page 1, line 23: replace 'avoid' with 'deal with'. Page 1, lines 16-19: rephrase ... (this is a bit weak, as first sentence of the article). Page 2, line 5: I suggest adding in this paragraph one sentence introducing the fourth class of spatial verification methods, the scale-separation techniques (with the key references). Then you start the new paragraph by stating that the technique introduced in your article belongs to this latter class. Then you describe the most recent literature on variograms etc. (as from line 8 onwards). Here you need to state that the variogram-based techniques are a sub-set of the scale-separation techniques. Page 2, the paragraph ending at line 22 can be joined with the one starting at line 23.

Section 2

Page 3 line 25: write 'The threshold T determines the percentage of the field which has non-zero values'. You need to state (here) that T is the base rate. Page 5, line 15: When introducing the scale auto-correlation parameter b, and when discussing Figure 1, you need to mention explicitly that smaller b are associated with larger scales, and vice-versa larger b are associated with smaller scales (this is counter-intuitive, therefore it needs to be reminded here and there in the article). Page 5, lines 11-13: it is not clear where this statement lead to: in the article, are you imposing $\nu > 1$? Are you using random Gaussian distributions to create / perturb you parameters? Please state. Page 5, line 26: define the rotation angle.

Section 3

Page 7, line14 - Page 8 line 1: this is not "loosely speaking", please redefine (in easier words) the concept of local stationarity: does it mean that locally your auto-correlation is zero? You can also decide to remain with mathematical strict definitions ... in the rest of the paragraph, you are quite technical ... however my preference is always to accompany the mathematical explanation with a sentence which explain / vulgarize the mathematical content. You might need to summarize the findings of Eckley et al (2010), Kapp et al. (2018).

Section 4

Re-title section 4 as 'Wavelet spectra spatial aggregation'. Page 9, line 10: for the case study add the reference to Ahijevych, D., E. Gilleland, B.G. Brown, and E.E. Ebert, 2009: Application of spatial verification methods to idealized and NWP-gridded precipitation forecasts. Weather Forecast., 24 (6), 1485 – 1497. [From the major comment: please, add a panel in Figure 3, with the mean spectra and histogram of the central scales for the shown case study. I expect a bimodal histogram and spectra, since both small scale features and a large front are present in the case study.]

Section 5

Re-title Section 5 as "Wavelet Spectra Sensitivity Analysis". Page 9, line 22: please remind here that larger (smaller) b is associated with smaller (larger) scales. Page 9, lines 26-27: eliminate the sentence "Simultaneously ... observed scales" (I do not see this in the Figure; moreover the sentence distracts from the main point).

Paragraph starting at page 9, line 32 and ending at page 10, line 2 (describing the major findings of Figure 4): In this paragraph you have one main result and a secondary result. The main result is that both mean spectra and scale histogram are sensitive to the variation in the parameters b and $\nu$, and that for both parameters they shift in the expected direction. The sensitivity of the spread as you vary b or $\nu$ is a secondary

results (which is actually neither too visible, nor to important for your study). In the paragraph these are mixed up in the discussion, so that the latter takes away the focus from the former. Rephrase the paragraph. E.g. at page 10, line 2, I suggest writing: ' ... only affected by b: larger scales (smaller b) lead to a greater variance (panel b) whereas changes in smoothness (parameter $\nu$) do not substantially change the histogram shape' (avoid mentioning the shift here). [From the major comment, you should also state that: 1. the scale histogram exhibits less spread, the dominant scales are better defined, and hence it is favoured wrt the mean spectra. 2. the smaller shift of the scale histogram is possibly proportional / due to its smaller spread, and not to a lack of sensitivity.]

Page 10, line 6: the lack of sensitivity of both the mean spectra and the scale histogram on the base rate (parameter T) is a very welcome property in a verification scoring rule (it implies that the score cannot be edged, e.g. by over-forecasting, and that the performance does not depend on the underlying climatology). This should be mentioned.

Section 6

Page 11: [From the major comment: real precipitation fields might generate bi-modal spectra (whereas the synthetic fields considered in the article have by construction uni-modal spectra). H_cd and Sp_cd (page 11), are not suitable statistics for comparing bi-modal (or non-Gaussian) spectra, because they compare the centre of mass of the curves: this limitation ought to be (at least) mentioned. H_emd and Sp_emd, on the other hand, seem more suitable statistics (to compare Gaussian or non-Gaussian curves) since based on the whole curve comparison. If the authors wish to introduce a metric which measure the direction of the error (such as H_cd and Sp_cd), maybe they should consider a measure based on the distances along the whole curves (or the integral between the two pdf), but with a sign which accounts for the curves relative position.]

Page Page 11, lines 10-13: please define EMD (either write the formula or describe how it is calculated ... "moving the dirt ... work" is visually clear, but it would be better to be more precise). Page 11, lines 13-14: by normalizing the spectra to obtain a unit sum you essentially remove the bias, and concentrate solely on the pure scale structure (how the total energy is distributed across the scales). This should be mentioned. Page 12, line 5: there is an incoherence in the naming of the Energy score, in this Section it is "Sp_e", whereas in Figure 5 it is "SpEn". I personally prefer the latter, or "Sp_en", to well separate it from "Sp_emd".

Section 7

Page 14, lines 11-13 (describing the bottom panels of Figure 5, evaluating the ensembles against a RS observation): not only the RS ensemble scores best (for all scores), but also the SmS and RL exhibits the second best score and the SmL (the most dissimilar ensemble with respect to RS) exhibits always the worst score. You should mention this.

From page 14 line 15, to page 15 line 6, need to be rephrased: a) when comparing RL to SmS (Page 14, bottom 2 lines): the compensating error affect solely the location / mean value of the mean spectra ans scale histogram, or does it affect the whole mean spectra and scale histogram? I question the phrasing 'on location of the spectra and histograms along the scale axis' (I would eliminate this part of the sentence). In the following sentence (page 15, line 1) I question 'by their centres of mass alone'. b) Page 15, lines 1-3: I think that the SmS and RL ensemble cannot be separated well for all scores (also Vw5), not only for Hcd (I won't attribute the lack of separation to the fact that Hcd compare centres of mass). This is possibly due to the fact that the mean spectra and scale histograms for RL and SmS are similar (From Figure 1, the top-left and bottom-right panels are more similar than the top-right versus bottom-left). Nevertheless, in the top panels of Figure 5 all scores (but V20) shows a slightly larger error for the SmS ensemble than for the RL ensemble (which is encouraging), and then even larger errors for SmL and RS (it seems to me that the scores are informative ...

). c) Top panels of Figure 5: The two scores considering the sign of the error (H_cd and S) exhibit the same behaviour, not only for SmL and RS, but also for SmS (they both exhibit slightly negative values): the sentences at page 15 lines 4-5 are partially incorrect, please re-phrase them.

From Figure 5 and 6, it is clear that V20 is the less informative score: please add this comment (you can relate to your comment when introducing V20 in Section 6, . . . ).

Section 7.2: The results associated to Figure 7 are very nicely discussed and very interesting! For ensembles, SpEn is the champion score followed by Vw5, whereas for deterministic Vw5 closely followed by Sp_emd are the champion scores. I am surprised of the lower performance of H_emd: why? After these results, one could be tempted to choose Vw5 as scoring rule . . . however its strong dependency on the base rate/climatology (Section 7.4) cannot be ignored. Maybe you can add some of this comment in the discussion?

Section 7.3: please specify in the caption of Table 3 (and Table 4), or write in the text, that ExP1 = D1 = Haar, and that Exp4 = D4 is the wavelet considered in the main experiment of the article.

Section 7.4, page 18 lines 5-6: given that in the original experiment T was set to 0.2 (aka 20% of the domain was precipitation, and 80% was zero values), I imagine that with this model the precipitation area is ranging in 15-25% of the domain: can you please phrase this more clearly? (rather than using the 75%-85% range, refer to your previously fix 20% base rate ... )

Discussion and conclusions

page 20, line 14: I suggest writing 'mis-representation of feature sizes (e.g. smoother representation of small-scale convective organization)'.

Page 20, lines 17-25: the findings of Figure 6 and 7 are well summarized in the conclusions (page 20, lines 21-25). I would end this paragraph at line 25. The sensitivity

of the Variogram score to p and w (lines 31-32) could also be added to this paragraph. Then (at page 20, line 26) I would start a new paragraph, discussing the results of the sensitivity analyses (sensitivity to T and to the wavelet choice).

Sensitivity to T: I suggest to phrase differently lines 25-30 (page 20): you need to remind that the 'perturbation of the data' is essentially an assessment of the sensitivity of the scores to the sample climatology. I would express more concern about the loss of discrimination of the variogram scores found in section 7.4.

Sensitivity to the wavelet choice: I would rephrase lines 32-33 (page 20) as 'We have also tested the sensitivity of the newly introduced wavelet-scores to the choice of the mother wavelet. We have performed . . . '.

As the last paragraph of the conclusion suggests, this study is still exploratory: there is no single score which has emerged as the recommended best score. This should be mentioned. Moreover, the paragraph could be re-phrased to include real case studies and scores which accounts for the direction of the error while applied to bi-modal spectra (as explained in the major comments).

---

## Referee Comment (RC2) · Joseph Bellier (Referee) · 24 May 2019

GENERAL COMMENTS:

This manuscript is the latest contribution to a field that has recently received attention in the verification literature: the use of wavelets for quantifying discrepancies in "texture" between high-resolution precipitation fields, as an alternative to verification tools that are affected by the double penalty (the classical point-wise measures), or to verification methods such as the object-based ones where discrimination capabilities can be found too dependent to intrinsic parameters. The authors expand the work presented in the previous related papers, by testing wavelet-based scores based on

different approaches for data reduction, but also by testing the discrimination capabilities of their scores on a controlled but realistic environment, leveraging an existing stochastic rainfall model.

Moreover, the authors investigate some aspects of the wavelet-based approach more in depth than in the previous papers, such as the choice of the mother wavelet, and the bias correction due the redundancy of the wavelet transform. I personally think that these latter investigations are equally interesting (compared to the development of the new scores), although the manuscript considers them as side results. I therefore encourage the authors to take the opportunity for expanding on these investigations, especially because I have found them missing in the previous related papers, and I am probably not the only one. I wouldn't require new experiments, but rather more in-depth discussions. See my specific comments below.

In overall, I have found the paper very well written, and clearly structured. The experiments are rigorous, the hypotheses are verified, and comparisons to existing alternatives in the literature are conducted. Moreover, the figures are of good quality.

However, I have found on several occasions that the introductions to the basic concepts of the approach were too brief. The consequence is that only a minority can read and understand the paper without having to read other related papers. I understand that the authors don't want to repeat things that are well explained elsewhere, but I would recommend a few more explanations on these concepts so that the paper is more self-sufficient.

All in all, I think this is a work of great quality, and I strongly recommend publication, pending minor corrections.

SPECIFIC COMMENTS:

P6 section 3: I have found the introduction to the wavelet theory pretty limited. I understand that you don't want to provide the mathematical details, but I think it is a

difficult start for a reader whose knowledges about wavelets is short. I would therefore recommend that you to start this section with a few sentences presenting (in words) what are wavelets, why are they popular for analyzing signals (or fields in the 2D case), and to which references the reader could refer for a more detailed (and mathematical) introduction.

P7 l5-6: I have two comments here. First of all, you may clarify that the squared weights quantify the energy spectra. Indeed, at this point of the paper you have used several times the term "spectra" but have not defined it, and you haven't used yet the term "energy". Second, I think it would be nice here to briefly mention the origin of this bias (the redundancy of the wavelet transforms?), and also which form does it takes (the energy increasing over and over with increasing scales?). In my opinion, how does this bias really affect the energy spectrum is something that has been poorly explained in the previous LS2W papers that you cite, and it would be nice to let the reader know what should he expect in case he doesn't apply the correction.

P7 l14-32: To my knowledge, your manuscript is the first (among the others having used the LS2W spectra for verifying precipitation fields) that investigates the choice of the mother wavelet. However, this paragraph is hard to grasp for a non-familiar reader, especially the differences between the wavelets. As a suggestion for improvement, I recommend that you add a figure of the plot of the different wavelets, so that it is easier to see the differences in terms of smoothness and support. You could also take the opportunity to refer to this figure at the very beginning of section 3, when introducing the mother wavelet function for the very first time. However, it is possible that the reader doesn't understand how one can apply a 1D function (the wavelet) to a 2D field, so it might be necessary to explain the process in few words (apply on the rows, then on the columns, etc.).

P8 l13-15: In my opinion, the fact that the amount of negative energy averages out if we choose a wavelet smoother than D1 should not be introduced as "Preliminary experiments have shown that . . .", but deserves a more detail paragraph and eventually supporting figures. Indeed, in my opinion, allowing negative energy is one of the biggest issue of the RDWT, so if you show that this problem vanishes by using other wavelets than the Haar wavelet, this is an important result, which should be discussed in more details.

P13 l15-19: It might it is necessary to give a little more explanation about the S component of the SAL (and perhaps a figure), so that your paper is self-sufficient. Moreover, you should say a few words about the ensemble version of SAL as well.

MINOR COMMENTS AND TECHNICAL CORRECTIONS:

p1 l19: "a given rain field is forecast perfectly, but slightly displaced": If there is a displacement error, then the field is not perfectly forecast. You may replace "field" by "object" or "feature".

p1 l23: After "four main strategies", the reader expects a descriptive list of each of these strategies. This is actually what you do in the paragraph that follows, but we have to wait until p2 l6 ("the last") to be sure that you are indeed referring to these four strategies. I recommend that you make the description more explicit.

p2 l17: remove the coma

p2 l22: You may briefly mention here the notion of "local stationarity".

p2 l25: As you write "corrected RDWT", you may later in the sentence say: "to obtain an unbiased estimate of the local wavelet spectra" (otherwise we don't know why you need to correct).

p2 l33: It is not clear why does considering both the ensemble and the deterministic case "avoid the need for further data reduction".

P7 l7: Isn't it "phi_{j,l,u}" instead of "phi_{j,j,u}"?

P7 l11: You say here that the smoothing is the final step of the spectra estimation procedure. However, in the package LS2W the smoothing takes place before the bias

correction by the matrix A-1. Please clarify.

P7 l25-26: I don't understand what do you mean by a "sparse representation". Please clarify.

P7 l28-30: You have defined at l23 the labels "ExP" and "LeA", but these are not used until P17. Maybe you could use them here (instead of "this version of D4").

P8 l9: It may be nice to remind why the invariance under shift is necessary.

P9 l6-7: Maybe it would be better to replace "(i,j)" (and elsewhere where you refer to the coordinates) by other indices such as "(x,y)", to avoiding confusion with J referring to scales.

P9 l11: Please indicate the rationale behind the logarithm transformation.

P9 l25: This introduction to Fig 4a is confusing, because you say "as a function of the scale parameter", but when you look at the Figure you read "scale" for the x-axis, although the scale parameter you are referring to is in the y-axis. Please clarify.

P10 Fig 4: For plots (a) and (b), you may change the style of the black dash line, as we actually don't see the dash.

P12 Equation (6): I'm wondering if readers unfamiliar with the energy score might be confused with your definition. Indeed, the name "Energy Score" has here nothing to do with the "energy" of the spectra you are referring to, and this might be confusing with the fact that you define y and F as the observed and forecast (energy) spectrum, although in the general definition of the energy score, y and F are simply the observation and the forecast, no matter which quantity is being forecast. A more general definition of the score may reduce the risk of confusion.

P12 l5: You never mention clearly in this paragraph that the forecast quantity at hand is a multivariate vector. Even if you bold the observation y and the realizations X and X', you should make crystal clear that it is a multivariate quantity, and give the dimension.

P12 Table 1: Actually, some of your scores (Hemd and Hcd) work for both deterministic and probabilistic forecasts, so maybe you could modify you table by either adding a column "deterministic" that you fill with "yes" or "no", or by modifying the title of your current column and fill it by "deterministic", "probabilistic" or "deterministic and proba-bilistic".

P13, Variogram score: It is not clear whether you apply the variogram score to quantities that represent the wavelet spectrum or the precipitation field. From p13 l5, I understand that X, y and EF refer to the vector of the spectrum, but later you refer to spatial locations, so that I figure out that your forecast quantities are fields, is that correct? Please clarify. More generally, please clarify which scores are built from the wavelet approach, and which ones from the precipitation fields directly.

P14 l12: I would add "for ensemble forecasts" (after "the established alternatives"), to clarify why you don't consider the RMSE here.

P14 and 15, Fig 5 and 6: the energy score is here referred to as SpEn, although in the text you refer to Spe. Similarly, in Fig 7 you refer to Semd, although in the text you refer to SPemd. In addition to these corrections, I think it would be nice to use the subscripts in the Figures, so that it is fully consistent with the text.
* * *

---

## Author Comment (AC1) · 27 Jun 2019

*We are grateful to all reviewers for their encouraging comments and extensive constructive criticism. Our answers to all individual points are given below. In the marked up manuscript, changes related to the comments of reviewer #1 are colored in magenta, those related to reviewer #2 are blue. The issue with the code availability raised by executive editor David Ham has been addressed as well.*

**Reviewer #1**

**Major Revisions:**

It would be nice to see the mean spectra and histogram of the dominant scales for the case study shown in Figure 3 (please add a panel): I expect a bimodal histogram and spectra, since both small scale features and a large front are present in the case study. This bi-modality (i.e. presence of of both small and large scale features) is not represented in the stochastic rain fields produced in Section 2 (the synthetic fields considered in the article have by construction uni-modal spectra), and in fact their resulting spectra and histograms are uni-modal (e.g. Figure 4). However spectra bimodality (i.e. presence of of both small and large scale features) is bound to happen in real verification practice, and it might be badly handled by H_cd and Sp_cd. In fact, H_cd and Sp_cd are the differences of the centre of mass (of the scale histograms and of the mean spectra), and are not suitable summary statistics to compare bimodal curves (or any other non-Gaussian curve). Hemd and Spemd, on the other hand, seem more suitable statistics (to compare Gaussian or non-Gaussian curves) since based on the whole curve comparison. The authors should consider withdrawing H_cd and Sp_cd from the newly proposed wavelet-based scores. [If the authors wish to introduce a metric which measure the direction of the error, maybe they should consider a measure based on the distances along the whole curves (or the integral between the two pdf), but with a sign which accounts for the curves relative position.]

Very good suggestion, we have added the mean spectrum and scale-histogram to the plot! As expected, the latter is indeed bi-modal. Since the two dominant scales are 5 and 6 (roughly), you cannot see two peaks in the mean spectrum - there is no notion of intermediate values between scales. We have, however, seen other real examples where both curves have multiple peaks.

We agree that Hcd and Spcd are not suitable to measure structure errors in a realistic setting. A comment to that effect has been added to the final section. We do however believe that the sign of these scores is usually a helpful quantity (noting that it typically agrees with SAL's S on the error's direction). We furthermore think it's worthwhile to actually demonstrate that the conceptually more complicated EMD is indeed needed and cannot simply be replaced by the difference in centre. Hcd and Spcd are therefore not completely 'withdrawn' from the study, but we have attempted to clarify their role as auxiliary quantities, mostly giving us a sign.

Figure 4 (Section 5) shows that both mean spectra and scale histograms are sensitive to the variation of the scale parameter b and the smoothness parameter v, and that for both parameters, the curves shift in the expected direction (this is the main result). The histogram of the dominant scales seems slightly less sensitive (it shifts less), however it exhibits a smaller spread (hence smaller uncertainty: the signal is better defined). Because of this latter property, the scale histogram should be favoured, with respect to the mean spectra. Moreover, the smaller shifts of the histograms are probably simply related / due to their smaller spread (I have the feeling that the magnitude of the shift is proportional to the spread). These aspects should be mentioned in Section 5. (Note: the sensitivity of the spread to the parameters b and v is secondary: be careful not to mix it up with the main result, aka the shift).

We agree to some extent that the shift is the main effect and the change in shape is secondary. To avoid mixing the two up, we have re-structured this part of the discussion such that the shift for both parameters and both curves is discussed first, then the change in shape is mentioned. The latter result is however relevant to the experiments presented later: If both nu and b only shifted the curves, we could not distinguish between the two kinds of errors.

The perceived "difference in spread" was actually due to the fact that the plots for histograms and spectra didn't have the same axes: The scale-axis for the histograms started at 0 even though 1 is the smallest possible central scale. After correcting the issue, the magnitude of the shift and the spread of the curves look more or less the same for histogram and spectrum. We would furthermore argue that one cannot simply compare the "spread" of these two directly anyways since they represent different mathematical entities with potentially very different degrees of freedom.

From the previous two comments, I would propose as unique new statistics H_emd.

At the end of Section 2, then authors introduce an algorithm for producing stochastic rain fields which satisfy non-stationarity and anisotropy. Some case studies are illustrated in Figure 2, and the associated verification results are discussed in Section 7.4. In my view this analysis can be removed from the article for the following reason: a) The algorithm for producing stochastic rain fields which satisfy non-stationarity and anisotropy, despite being more sophisticated than the isotropic algorithm mainly used in the article, is still not realistic (the precipitation features of Figure 2 are still far from resembling the ones for the real case illustrated in Figure 3). b) The article will result nicely well contained in illustrating "solely" the isotropic stochastic fields (you have already quite a lot of material! Moreover, this would provide a nice "excuse" for retaining the statistics based on the centres of mass -wink!-). In this case you need to add into the final discussion Section the need to analyze real cases, in
future work ...
c) For the (future) analysis of more realistic cases, I strongly suggest to consider directly real precipitation case studies (the Spatial Verification ICP cases from Ahijevych et al 2009 are available online), rather than using synthetic fields (you might end up spending

a lot of time and implementing very complex stochastic models ... to achieve the same results ... ).

After some deliberation, we have decided to follow your argumentation and **remove the non-stationary example from the paper**. We agree that the additional results from this experiment are outweighed by the advantages of having a shorter more streamlined paper, especially since the realism of our non-stationary model is indeed questionable.

**Minor Revisions:**
Abstract and Introduction

Page 1 line 7: replace 'spatial correlation' with 'spatial structure' (or 'scale structure').
changed it.

Page 1, line 23: please quote (also) Dorninger et al (2018): "The set-up of the Mesoscale Verification Inter-Comparison over Complex Terrain project". Bull. Amer. Meteorol. Soc., 99 (9), 1887 – 1906.
Added the reference. Now the newly discovered fifth verification class is also briefly mentioned.

Page 1, line 23: replace 'avoid' with 'deal with'.
ok

Page 1, lines 16-19: rephrase… (this is a bit weak, as first sentence of the article).
The first few sentences have been replaced by a (hopefully) less lame introduction.

Page 2, line 5: I suggest adding in this paragraph one sentence introducing the fourth class of spatial verification methods, the scale-separation techniques (with the key references). Then you start the new paragraph by stating that the technique introduced in your article belongs to this latter class. Then you describe the most recent literature on variograms etc. (as from line 8 onwards). Here you need to state that the variogram-based techniques are a sub-set of the scale-separation techniques.
Re-structured the two paragraphs accordingly.

Page 2, the paragraph ending at line 22 can be joined with the one starting at line 23.
The two have been joined.

Section 2
Page 3 line 25: write 'The threshold T determines the percentage of the field which has non-zero values'. You need to state (here) that T is the base rate.
We have added that clarification.

Page 5, line 15:
When introducing the scale auto-correlation parameter b, and when discussing Figure 1,

you need to mention explicitly that smaller b are associated with larger scales, and vice-versa larger b are associated with smaller scales (this is counter-intuitive, therefore it needs to be reminded here and there in the article).

Tried to make this more explicit.

Page 5, lines 11-13: it is not clear where this statement lead to: in the article, are you imposing v > 1? Are you using random Gaussian distributions to create / perturb you parameters? Please state.

Since the model contains second derivatives of the Matern fields, it will crash if v <=1, so we only used v > 1. This technical detail is not actually necessary to understand the rest of the paper, so we simply cut it.

Page 5, line 26: define the rotation angle.

Obsolete since we cut the nonstationary model.

Section 3
Page 7, line 14 - Page 8 line 1: this is not "loosely speaking", please redefine (in easier words) the concept of local stationarity: does it mean that locally your auto-correlation is zero? You can also decide to remain with mathematical strict definitions ... in the rest of the paragraph, you are quite technical ... however my preference is always to accompany the mathematical explanation with a sentence which explain / vulgarize the mathematical content. You might need to summarize the findings of Eckley et al (2010), Kapp et al. (2018).

You are correct, the way we speak here is not particularly loose (that formulation has been cut). We believe that further technical details about the nature of local stationary would distract from the main point - for the paper it is sufficient to know that correlations can vary in space as long as they do so slowly. In fact, neither Eckley nor Kapp give a general definition of local stationarity, they only state the specific regularity conditions imposed on the LS2W.

Section 4
Re-title section 4 as 'Wavelet spectra spatial aggregation'.

As you wish.

Page 9, line 10: for the case study add the reference to Ahijevych, D., E. Gilleland, B.G. Brown, and E.E. Ebert, 2009: Application of spatial verification methods to idealized and NWP-gridded precipitation forecasts. Weather Forecast., 24 (6), 1485 – 1497.

We moved the reference from the figure caption to the body of the text.

[From the major comment: please, add a panel in Figure 3, with the mean spectra and histogram of the central scales for the shown case study. I expect a bimodal histogram and spectra, since both small scale features and a large front are present in the case study.]

**Section 5**

Re-title Section 5 as "Wavelet Spectra Sensitivity Analysis".
Done.

Page 9, line 22: please remind here that larger (smaller) b is associated with smaller (larger) scales. Page 9, lines 26-27: eliminate the sentence "Simultaneously ... observed scales" (I do not see this in the Figure; moreover the sentence distracts from the main point). Paragraph starting at page 9, line 32 and ending at page 10, line 2 (describing the major findings of Figure 4): In this paragraph you have one main result and a secondary result. The main result is that both mean spectra and scale histogram are sensitive to the variation in the parameters b and v, and that for both parameters they shift in the expected direction. The sensitivity of the spread as you vary b or v is a secondary results (which is actually neither too visible, nor to important for your study). In the paragraph these are mixed up in the discussion, so that the latter takes away the focus from the former. Rephrase the paragraph. E.g. at page 10, line 2, I suggest writing: ' … only affected by b: larger scales (smaller b) lead to a greater variance (panel b) whereas changes in smoothness (parameter v) do not substantially change the histogram shape' (avoid mentioning the shift here). [From the major comment, you should also state that: 1. the scale histogram exhibits less spread, the dominant scales are better defined, and hence it is favoured wrt the mean spectra. 2. the smaller shift of the scale histogram is possibly proportional / due to its smaller spread, and not to a lack of sensitivity.]
See answer to the major comment.

Page 10, line 6: the lack of sensitivity of both the mean spectra and the scale histogram on the base rate (parameter T) is a very welcome property in a verification scoring rule (it implies that the score cannot be edged, e.g. by over-forecasting, and that the performance does not depend on the underlying climatology). This should be mentioned.
We agree that it should be mentioned, but feel that such judgemental statements should better be relegated to the discussion at the end of the paper.

**Section 6**

Page 11: [From the major comment: real precipitation fields might generate bi-modal spectra (whereas the synthetic fields considered in the article have by construction uni-modal spectra). H_cd and Sp_cd (page 11), are not suitable statistics for comparing bi-modal (or non-Gaussian) spectra, because they compare the centre of mass of the curves: this limitation ought to be (at least) mentioned. H_emd and Sp_emd, on the other hand, seem more suitable statistics (to compare Gaussian or non-Gaussian curves) since based on the whole curve comparison. If the authors wish to introduce a metric which measure the direction of the error (such as H_cd and Sp_cd), maybe they should consider a measure based on the distances along the whole curves (or the integral between the two pdf), but with a sign which accounts for the curves relative position.]

See answer to the major comment above.

Page 11, lines 10-13: please define EMD (either write the formula or describe how it is calculated ... "moving the dirt ... work" is visually clear, but it would be better to be more precise).

We have added a clarification of what exactly corresponds to the mass/location of the dirt piles. We furthermore mention a simple way of calculating the relevant special case of the EMD without numerical optimization (the simplification has only recently come to our attention).

Page 11, lines 13-14: by normalizing the spectra to obtain a unit sum you essentially remove the bias, and concentrate solely on the pure scale structure (how the total energy is distributed across the scales). This should be mentioned.

Good point, we now mention that.

Page 12, line 5: there is an incoherence in the naming of the Energy score, in this Section it is "Sp_e", whereas in Figure 5 it is "SpEn". I personally prefer the latter, or "Sp_en", to well separate it from "Sp_emd".

We agree, changed it to Sp_en.

Section 7

Page 14, lines 11-13 (describing the bottom panels of Figure 5, evaluating the ensembles against a RS observation): not only the RS ensemble scores best (for all scores), but also the SmS and RL exhibits the second best score and the SmL (the most dissimilar ensemble with respect to RS) exhibits always the worst score. You should mention this.

Good idea, we have added that observation.

From page 14 line 15, to page 15 line 6, need to be rephrased:
a) when comparing RL to SmS (Page 14, bottom 2 lines): the compensating error affect solely the location /mean value of the mean spectra and scale histogram, or does it affect the whole mean spectra and scale histogram? I question the phrasing 'on location of the spectra and histograms along the scale axis' (I would eliminate this part of the sentence). In the following sentence (page 15, line 1) I question 'by their centres of mass alone'.

You are correct in that the two changed parameters affect both location and shape of the curve, we have re-ordered the relevant sentence to make it clear that we don't mean that the shape is unaffected. The effect on the shape of the curves, however, has (to first order) the same sign: An increase in smoothness and an increase in b (decrease in scale) both lead to distributions which are less spread out. These kinds of errors *do not* compensate each other, the histogram of SmS is unambiguously too tight. We therefore stand by our statement that the centre alone is insufficient but the EMD can, in principle distinguish the two.

b) Page 15, lines 1-3: I think that the SmS and RL ensemble cannot be separated well for all scores (also Vw5), not only for Hcd (I won't attribute the lack of separation to the fact that Hcd compare centres of mass). This is possibly due to the fact that the mean spectra and scale histograms for RL and SmS are similar (From Figure 1, the top-left and bottom-right panels are more similar than the top-right versus bottom-left). Nevertheless, in the top panels of Figure 5 all scores (but V20) shows a slightly larger error for the SmS ensemble than for the RL ensemble (which is encouraging), and then even larger errors for SmL and RS (it seems to me that the scores are informative ...).

You are correct, the spectra are certainly more similar than those compared in the first half of the experiment. As discussed in the answer to the previous comment, we stand by our assertion that there is a well defined difference between the two curves which can, in principle, be measured by EMD but not cd.

c) Top panels of Figure 5: The two scores considering the sign of the error (H_cd and S) exhibit the same behaviour, not only for SmL and RS, but also for SmS (they both exhibit slightly negative values): the sentences at page 15 lines 4-5 are partially incorrect, please re-phrase them.

It has been re-phrased to note that the tendency is the same, but we stand by the claim that the signal is slightly stronger for S.

From Figure 5 and 6, it is clear that V20 is the less informative score: please add this comment (you can relate to your comment when introducing V20 in Section 6, ...).

That's right, but we feel that the unsurprisingly meagre performance of V20 is sufficiently mentioned in the final section and requires no additional discussion.

Section 7.2: The results associated to Figure 7 are very nicely discussed and very interesting! For ensembles, SpEn is the champion score followed by Vw5, whereas for deterministic Vw5 closely followed by Sp_emd are the champion scores. I am surprised of the lower performance of H_emd: why? After these results, one could be tempted to choose Vw5 as scoring rule … however its strong dependency on the base rate/climatology (Section 7.4) cannot be ignored. Maybe you can add some of this comment in the discussion?

We have attempted to clarify in the discussion that dependence on the base rate is undesirable for a pure structure score. We could however imagine verification settings where one wants to judge structure and precipitation area simultaneously. We therefore refrain from calling the wavelet-based score 'better'.

Section 7.3: please specify in the caption of Table 3 (and Table 4), or write in the text, that ExP1 = D1 = Haar, and that Exp4 = D4 is the wavelet considered in the main experiment of the article.

Good idea to remind the reader which wavelet we've been using. We have added that to the caption of Table 3 (Tab. 4's caption is just "as Tab. 3, but ...").

Section 7.4, page 18 lines 5-6: given that in the original experiment T was set to 0.2 (aka

20% of the domain was precipitation, and 80% was zero values), I imagine that with this model the precipitation area is ranging in 15-25% of the domain: can you please phrase this more clearly? (rather than using the 75%-85% range, refer to your previously fix 20% base rate ... )

That was actually just a typo, it should of course read "a uniform random variable between 15 % and 25 % of the complete domain", not 75%-85%. Fixed it.

Discussion and conclusions

page 20, line 14: I suggest writing 'mis-representation of feature sizes (e.g. smoother representation of small-scale convective organization)'.

Changed it accordingly. The previously given example of "missing fronts" is indeed a bit misleading since that is typically a matter of displacement, not structure.

Page 20, lines 17-25: the findings of Figure 6 and 7 are well summarized in the conclusions (page 20, lines 21-25). I would end this paragraph at line 25. The sensitivity of the Variogram score to p and w (lines 31-32) could also be added to this paragraph. Then (at page 20, line 26) I would start a new paragraph, discussing the results of the sensitivity analyses (sensitivity to T and to the wavelet choice).

The sensitivity-experiments are now in a new paragraph. The variogram score's sensitivity was not moved because the paragraph previously ending at line 25 now also contains a remark regarding the possibility of multi-modal spectra (see answer above) and would have been too full.

Sensitivity to T: I suggest to phrase differently lines 25-30 (page 20): you need to remind that the 'perturbation of the data' is essentially an assessment of the sensitivity of the scores to the sample climatology. I would express more concern about the loss of discrimination of the variogram scores found in section 7.4.

The remark about the sample climatology has been added. As discussed in the answer to the comment concerning 7.2, we feel that an appropriate amount of concern has been expressed - the two approaches measure different things and the wavelet-scores isolate structural characteristics more clearly.

Sensitivity to the wavelet choice: I would rephrase lines 32-33 (page 20) as 'We have also tested the sensitivity of the newly introduced wavelet-scores to the choice of the mother wavelet. We have performed ...'.

They have been rephrased, albeit with slightly different grammar to avoid the double "We have ...".

As the last paragraph of the conclusion suggests, this study is still exploratory: there is no single score which has emerged as the recommended best score. This should be mentioned. Moreover, the paragraph could be re-phrased to include real case studies and scores which accounts for the direction of the error while applied to bi-modal spectra (as explained in the major comments).

We have slightly re-phrased the beginning of this paragraph to make it clear that the first study which applies those scores to the real world will still be experimental in nature. Whether or not there is a single best score is not really the topic of this paper. Early results from our next study, however, suggest that the difference between the two score-families is actually pretty small in real-world situations. In that case hEMD seems preferable to us, but that is a story for another time ...

**Reviewer # 2**

**SPECIFIC COMMENTS:**

P6 section 3: I have found the introduction to the wavelet theory pretty limited. I understand that you don't want to provide the mathematical details, but I think it is a difficult start for a reader whose knowledges about wavelets is short. I would therefore recommend that you to start this section with a few sentences presenting (in words) what are wavelets, why are they popular for analyzing signals (or fields in the 2D case), and to which references the reader could refer for a more detailed (and mathematical) introduction.

*Fair point, we have added a few more words and literature recommendations.*

P7 l5-6: I have two comments here. First of all, you may clarify that the squared weights quantify the energy spectra. Indeed, at this point of the paper you have used several times the term "spectra" but have not defined it, and you haven't used yet the term "energy".

*Good idea, the term "wavelet spectrum" is now introduced at this point.*

Second, I think it would be nice here to briefly mention the origin of this bias (the redundancy of the wavelet transforms?), and also which form does it takes (the energy increasing over and over with increasing scales?). In my opinion, how does this bias really affect the energy spectrum is something that has been poorly explained in the previous LS2W papers that you cite, and it would be nice to let the reader know what should he expect in case he doesn't apply the correction.

*Yes, that would be nice. We have added a few words to that effect.*

P7 l14-32: To my knowledge, your manuscript is the first (among the others having used the LS2W spectra for verifying precipitation fields) that investigates the choice of the mother wavelet. However, this paragraph is hard to grasp for a non-familiar reader, especially the differences between the wavelets. As a suggestion for improvement, I recommend that you add a figure of the plot of the different wavelets, so that it is easier to see the differences in terms of smoothness and support. You could also take the opportunity to refer to this figure at the very beginning of section 3, when introducing the mother wavelet function for the very first time. However, it is possible that the reader doesn't understand how one can apply a 1D function (the wavelet) to a 2D field, so it might be necessary to explain the process in few words (apply on the rows, then on the columns, etc.).

*We have added a small figure, showing the 2D-version of the first four Daubechies wavelets. This hopefully gives the reader a better idea of the different basis functions and doesn't confuse them with the problem of applying the 1D mother to a 2D field: In principle, one could also calculate each coefficient by individually multiplying the field with the 2D daughter wavelet. This would be terribly inefficient, but for the purposes of this paper we don't really need to worry about the technical implementation of the RDWT.*

P8 l13-15: In my opinion, the fact that the amount of negative energy averages out if we choose a wavelet smoother than D1 should not be introduced as "Preliminary experiments have shown that . . .", but deserves a more detail paragraph and eventually supporting figures. Indeed, in my opinion, allowing negative energy is one of the biggest issue of the RDWT, so if you show that this problem vanishes by using other wavelets than the Haar wavelet, this is an important result, which should be discussed in more details.

We agree that this observation is not altogether unimportant. A figure showing the actual ratios between negative and total energy, as well as a few explanatory words, have been added to the appendix. Since the asymptotic theory of locally stationary processes is not, however, the focus of this study, we feel that more in-depth discussion of negative energy would distract readers from the core points of the paper. As you said in previous comments, the material is already not trivial if one has never worked with wavelets before. For our purposes it suffices to know that smooth wavelets mostly eliminate the problem.

P13 l15-19: It might it is necessary to give a little more explanation about the S component of the SAL (and perhaps a figure), so that your paper is self-sufficient. Moreover, you should say a few words about the ensemble version of SAL as well.

You're probably right, it is better to have a short explanation of how S measures structure so that readers don't need to look at a second paper just to understand this part of the experiment. A few words have been added.

**MINOR COMMENTS AND TECHNICAL CORRECTIONS:**

p1 l19: "a given rain field is forecast perfectly, but slightly displaced": If there is a displacement error, then the field is not perfectly forecast. You may replace "field" by "object" or "feature".

Done.

p1 l23: After "four main strategies", the reader expects a descriptive list of each of these strategies. This is actually what you do in the paragraph that follows, but we have to wait until p2 l6 ("the last") to be sure that you are indeed referring to these four strategies. I recommend that you make the description more explicit.

The section has been re-formulated slightly to make it clear that we have begun counting to four.

p2 l17: remove the coma

It is gone.

p2 l22: You may briefly mention here the notion of "local stationarity".

Good idea.

p2 l25: As you write "corrected RDWT", you may later in the sentence say: "to obtain an

unbiased estimate of the local wavelet spectra" (otherwise we don't know why you need to correct).

Added that remark.

p2 l33: It is not clear why does considering both the ensemble and the deterministic case "avoid the need for further data reduction".

It does not, those are just two separate things we do differently from Kapp 2018. The sentence has been clarified.

P7 l7: Isn't it "phi_{j,l,u}" instead of "phi_{j,j,u}"?

Fixed the typo.

P7 l11: You say here that the smoothing is the final step of the spectra estimation procedure. However, in the package LS2W the smoothing takes place before the bias correction by the matrix A-1. Please clarify.

Yes, Eckley does the smoothing prior to bias correction because the distribution of the uncorrected spectra should be chi-squared and we know how to smooth chi-squared variables with wavelet shrinkage. The formulation has been changed.

P7 l25-26: I don't understand what do you mean by a "sparse representation". Please clarify.

It just means that most coefficients are nearly zero. We have clarified that.

P7 l28-30: You have defined at l23 the labels "ExP" and "LeA", but these are not used until P17. Maybe you could use them here (instead of "this version of D4").

Good idea.

P8 l9: It may be nice to remind why the invariance under shift is necessary.

We have added a comment to that effect.

P9 l6-7: Maybe it would be better to replace "(i,j)" (and elsewhere where you refer to the coordinates) by other indices such as "(x,y)", to avoiding confusion with J referring to scales.

Good point, changed it to (x,y). Changed the indices in the section about the vg-score as well.

P9 l11: Please indicate the rationale behind the logarithm transformation.

Whether or not it is a good idea to log-transform real rain fields prior to the wavelet-transform is an interesting question which we plan to discuss in some detail in the next publication. Since it is really only necessary for this one example image, we feel that a full discussion is not warranted at this point. We have added a short remark stating that the log-transform leads to a more well-behaved marginal distribution and reduces the impact of extremes: If you leave the data as it is, the wavelet spectrum can be dominated by singular extreme pixels while large regions have virtually no influence. That is at odds with our intuitive idea of "structure".

P9 l25: This introduction to Fig 4a is confusing, because you say "as a function of the scale parameter", but when you look at the Figure you read "scale" for the x-axis, although the scale parameter you are referring to is in the y-axis. Please clarify.
We have attempted to clarify.

P10 Fig 4: For plots (a) and (b), you may change the style of the black dash line, as we actually don't see the dash.
We have attempted to make the dashed line a little more dashed, but the plot3D-library is a stubborn beast. A clarification has been added to the caption to make sure that everyone knows which lines we mean.

P12 Equation (6): I'm wondering if readers unfamiliar with the energy score might be confused with your definition. Indeed, the name "Energy Score" has here nothing to do with the "energy" of the spectra you are referring to, and this might be confusing with the fact that you define y and F as the observed and forecast (energy) spectrum, although in the general definition of the energy score, y and F are simply the observation and the forecast, no matter which quantity is being forecast. A more general definition of the score may reduce the risk of confusion.
Good point, we made it seem like the original definition of the energy score is somehow related to wavelets. The sentence has been clarified.

P12 l5: You never mention clearly in this paragraph that the forecast quantity at hand is a multivariate vector. Even if you bold the observation y and the realizations X and X', you should make crystal clear that it is a multivariate quantity, and give the dimension.
We have added the words "multivariate" and "vector" to bring that point across.

P12 Table 1: Actually, some of your scores (Hemd and Hcd) work for both deterministic and probabilistic forecasts, so maybe you could modify you table by either adding a column "deterministic" that you fill with "yes" or "no", or by modifying the title of your current column and fill it by "deterministic", "probabilistic" or "deterministic and probabilistic".
A further column has been added.

P13, Variogram score: It is not clear whether you apply the variogram score to quantities that represent the wavelet spectrum or the precipitation field. From p13 l5, I understand that X, y and EF refer to the vector of the spectrum, but later you refer to spatial locations, so that I figure out that your forecast quantities are fields, is that correct? Please clarify. More generally, please clarify which scores are built from the wavelet approach, and which ones from the precipitation fields directly.
That was in fact a mistake, X,y and F are not at all the same as in equation 6! We corrected that and made sure to mention at the beginning of the section that the following scores are non-wavelet alternatives. The division into wavelet and non-wavelet is also reflected in table

1.

P14 l12: I would add "for ensemble forecasts" (after "the established alternatives"), to clarify why you don't consider the RMSE here.
It has been added.

P14 and 15, Fig 5 and 6: the energy score is here referred to as SpEn, although in the text you refer to Spe. Similarly, in Fig 7 you refer to Semd, although in the text you refer to SPemd. In addition to these corrections, I think it would be nice to use the subscripts in the Figures, so that it is fully consistent with the text.
The notation has been unified, the plots now also have the subscripts.

**Executive editor comment (David Ham)**

am writing as an executive editor of GMD to highlight an issue with the code availability section which needs to be remedied in the revised manuscript. Thank you for providing a reference to the full code and data used in the experiments presented in your manuscript. There are two problems with providing this data via GitHub. The first is that a reader cannot identify the exact version of the code that was used in the paper (for example, you may fix bugs or add features in the future). The second issue is that projects sometimes change the revision control system they use, or the hosting (the project might move to GitLab, for example). The solution to both of these issues is to provide a reference to a persistent archive of the exact version of the code that was used in the manuscript. This reference can, and should, be in addition to the GitHub link, so that a user can also always access the most recent version of the code.

Since your original code is hosted on GitHub, the easiest way to produce a persistent archive of a precise version is to use GitHub's Zenodo integration. For more details, see: https://guides.github.com/activities/citable-code/. Please ensure that the revised version of your manuscript contains a reference to a persistent, public archive of the exact version of the code used to produce it.

Thank you for the recommendation, we have done as you said and uploaded the version used in the paper to Zenodo: 10.5281/zenodo.3257511

[revised manuscript text omitted]